*Method*

# Unraveling mitotic protein networks by 3D multiplexed epitope drug screening

Lorenz J Maier[1,2,3,†] (iD), Stefan M Kallenberger[1,2,3,†], Katharina Jechow[2,4], Marcel Waschow[2], Roland Eils[1,2,4,*] (iD) & Christian Conrad[1,2,**] (iD)

## Abstract

Three-dimensional protein localization intricately determines the functional coordination of cellular processes. The complex spatial context of protein landscape has been assessed by multiplexed immunofluorescent staining or mass spectrometry, applied to 2D cell culture with limited physiological relevance or tissue sections. Here, we present 3D SPECS, an automated technology for 3D Spatial characterization of Protein Expression Changes by microscopic Screening. This workflow comprises iterative antibody staining, high-content 3D imaging, and machine learning for detection of mitoses. This is followed by mapping of spatial protein localization into a spherical, cellular coordinate system, a basis for model-based prediction of spatially resolved affinities of proteins. As a proof-of-concept, we mapped twelve epitopes in 3D-cultured spheroids and investigated the network effects of twelve mitotic cancer drugs. Our approach reveals novel insights into spindle fragility and chromatin stress, and predicts unknown interactions between proteins in specific mitotic pathways. 3D SPECS's ability to map potential drug targets by multiplexed immunofluorescence in 3D cell culture combined with our automated high-content assay will inspire future functional protein expression and drug assays.

**Keywords** cell profiling; mitosis modeling; multiplexed immunostaining; protein–protein interactions

**Subject Categories** Cell Cycle; Genome-Scale & Integrative Biology; Methods & Resources

**Mol Syst Biol. (2018) 14: e8238**

## Introduction

Cellular processes are inherently linked to changes in spatial distributions of proteins. These alterations of distribution patterns are a result of dynamic interactions between proteins, affinities to cellular compartments, or the formation of functional protein complexes. Multiplexed immunostaining (Gerdes *et al*, 2013; Lin, 2017), in combination with 3D microscopy, extends the amount of information about spatial protein distributions that can be extracted for studying cellular systems. In particular in complex cellular events, simultaneously measuring distributions of different proteins in the same cell allows studying functional relations between proteins, or even functional complexes of several proteins. Potentially, all components of whole signaling pathways can be simultaneously studied. However, integrated methods for automation, segmentation, and efficient multichannel evaluation are not presently combined or available. Moreover, obstacles for systematically investigating the architecture of cellular processes result from topological cell-to-cell variability, whereas cells self-organize, re-structure, and are differently orientated, hampering a direct comparative analysis of different data sets containing groups of individual cells or spheroids.

Here, we present an automated technology for 3D Spatial characterization of Protein Expression Changes by microscopic Screening (3D SPECS) which facilitates sequential immunostaining using a pipetting robot, automated high-content image acquisition, and multichannel analysis. For evaluating 3D image data of topographically structured cellular events in standardized maps, we established a novel segmentation-free representation named SpheriCell that makes use of a protein-specific landmark-based registration. This multichannel registration is the prerequisite for systematic comparisons between groups of cells in different states or for evaluating effects of drugs on the intracellular topography analysis. It is subsequently linked to mathematical models of intracellular biochemical processes. Hence, our approach allows to assess co-localization affinities of proteins and their preferred localizations within intracellular maps consisting of spherical ROIs.

Establishing this workflow was initially motivated by the goal of applying multiplexed staining to drug screening, to add an additional information-rich layer of inhibitory processes in mitotic pathways, as several drug compounds targeting cell division unexpectedly failed in clinical trials (Chan *et al*, 2012; Marques *et al*,

1 Center for Quantitative Analysis of Molecular and Cellular Biosystems (BioQuant), Heidelberg University, Heidelberg, Germany
2 Division of Theoretical Bioinformatics, German Cancer Research Center (DKFZ), Heidelberg, Germany
3 Department for Bioinformatics and Functional Genomics, Institute for Pharmacy and Molecular Biotechnology (IPMB), Heidelberg University, Heidelberg, Germany
4 BIH Center for Digital Health, Charité Universitätsmedizin Berlin and Berlin Institute of Health, Berlin, Germany
 *Corresponding author. Tel: +49 30 450 543088; E-mail: r.eils@dkfz.de
 **Corresponding author. Tel: +49 6221 54 51304; E-mail: c.conrad@dkfz.de
 †These authors contributed equally to this work

2015; Otto & Sicinski, 2017). Conventionally, drug screens only account for measures as IC50 values in monolayer or, more recently, in 3D cell cultures (Jabs et al, 2017). We applied our 3D SPECS workflow to quantitatively study the differential topography of mitosis in tumorigenic MCF10CA and non-tumorigenic MCF10A cells. Experiments were performed in a 3D cell culture system regarded as physiologically more relevant than a planar cell culture because several features of these cells as differentiation, growth arrest, or formation of acinar structures depend on 3D growth (Imbalzano et al, 2009). To capture the most critical events, we distinguished between cells in metaphase or cells during segregation (anaphase and telophase combined). We chose this well-established model of a non-malignant progenitor and in vitro-derived malignant cells to sensitively characterize phenotypic changes in the cellular architecture during mitosis related to malignant transformation. Especially, we probed drug effects, in a multiplexed manner, on mitotic spindle organization, spindle assembly checkpoint (SAC), and complementary cell fate indicators. The SAC control includes the chromosome passenger complex (CPC) comprising BIRC5, Borealin, INCENP, and Aurora B (Carmena et al, 2012), which inhibit the segregating anaphase promoting complex (APC/C) most efficiently through mitotic checkpoint complex (MCC) containing among others BUB1β (BUBR1; London & Biggins, 2014). Failures in these specific mitotic checkpoints can lead to disruption and catastrophe of mitosis followed by autophagic or necrotic events and therefore are investigated as potential anti-cancer drugs.

The success of future mitotic checkpoint-targeted cancer therapies will depend on such complex 3D cell culture-based screens to uncover synthetic lethal interaction or resistance of potent compounds in vulnerable mitotic cancer cells (Chan et al, 2012; Otto & Sicinski, 2017).

# Results

### Applying 3D SPECS for mapping protein distributions in topographically ordered cellular events

Iterative antibody labeling overcomes the spectral limit of total number of fluorescent antibodies that can be applied simultaneously to individual cells (Gerdes et al, 2013; Lin, 2017). We extended this technique of chemically bleached fluorescently labeled antibodies, to 3D cell cultured spheroids in Matrigel (Petersen et al, 1992), combined with twelve different drug treatments (see Appendix Table S1). Our setup (Fig 1A) uses confocal laser scanning microscopy together with automated detection of mitoses by machine learning, and a motorized in-built micropipetting robot to comprehensively stain mitotic phases.

Usually, mitotic cells in culture divide into different orientations, which complicates comparisons between different sets of single-cell data. To investigate mitosis as an example for a topographically ordered cellular process, we applied a novel representation named SpheriCell that facilitates spatial alignment of subcellular events by registration of a spherical coordinate to cellular landmarks. Within the defined spherical coordinate systems of the cellular space, protein concentrations are then measured in a standardized set of 3D partitions.

For cells in metaphase, the spindle axis perpendicular to the metaphase plate was used as landmark (Fig 1B). The mitotic axis was defined by the shortest half axis of an ellipsoid fitted to the nuclear DAPI signal. Next, three sectors were delineated relative to the mitotic axis, either parallel (polar), diagonal, or in the division plane (equatorial). Six shells with equal radius intervals were centered to the nuclear ellipsoid in a way that the fourth shell was scaled to the longest half axis of the ellipsoid (Fig EV1A). For cells during segregation, the mitotic axis was specified by the line between two ellipsoids fitted to the daughter nuclei (Fig EV1B). Six equally spaced shells were defined by centering the fourth shell to the centers of the two ellipsoids. Following this procedure for cells in metaphase and segregation, outlines of cells growing in spheroids were approximated. We chose the size of the SpheriCell maps to fully cover intracellular protein distributions of the observed proteins involved in mitosis. Finally, a system of 18 spherical ROIs was created by intersecting sectors and shells. SpheriCell maps were projected on 2D planes for enhanced visualization (Fig 1B).

Maps of spherical ROIs were created for 12 proteins involved in mitosis, measured by iterative staining, and the DAPI signal, either in MCF10A or in MCF10CA cells. Measurements were recorded in cells treated with one of 12 different inhibitors or in untreated cells. In total, we screened 6,272 confocal image stacks and recorded 1,217 mitotic events resulting in 284,778 mean intensity values of 3D partitions.

The applied human epithelial MCF10 breast cancer progression model compares the near-diploid non-malignant cell line MCF10A forming polarized spheroids (Debnath & Brugge, 2005) with the tumorigenic invasively growing line MCF10CA, which bear activating mutations of HRAS and PIK3CA, and amplified MYC (Maguire et al, 2016). At first, we confirmed known localizations of cellular proteins and known mitotic checkpoints for untreated cells as described before including β- and γ-tubulin, γ-H2AX, Aurora kinases, SAC, and CPC complexes (Nogales et al, 1998; Rogakou et al, 1998; Carmena et al, 2012), supporting the utility of 3D SPECS (Figs 1C and 2A).

MCF10CA staining patterns resembled those of MCF10A showing a slightly reduced average DAPI signal due to increased size of MCF10CA nuclei (Fig 2B). Highest protein concentration increases were observed for γ-H2AX and Aurora A, contrasted by a reduction strongest for γ-tubulin. Increased levels of γ-H2AX (Paull et al, 2000), a marker for double-strand breaks, most likely reflect higher chromosomal stress in MCF10CA. Higher intensity levels of Aurora A, which is upregulated during mitosis and localizes mostly toward centrosomes (Carmena & Earnshaw, 2003), are consistent with previously described effects upon activation of Raf-1, downstream of the oncogenic RAS pathway (D'Assoro et al, 2014).

Taken together, the 3D SPECS approach was established to quantitatively study intracellular protein distributions during topographically structured cellular processes by 3D registration and fitting a spherical coordinate system to microscopy data of an ordered cellular event. Subsequently, the spherical coordinate system is subdivided into sectors and shells, and projected to a two-dimensional map. We quantitatively studied localization patterns of proteins involved in mitosis and observed distinctive differences between malignant and non-malignant cells, which serves as basis for characterizing the influences of drugs on intracellular protein distributions.

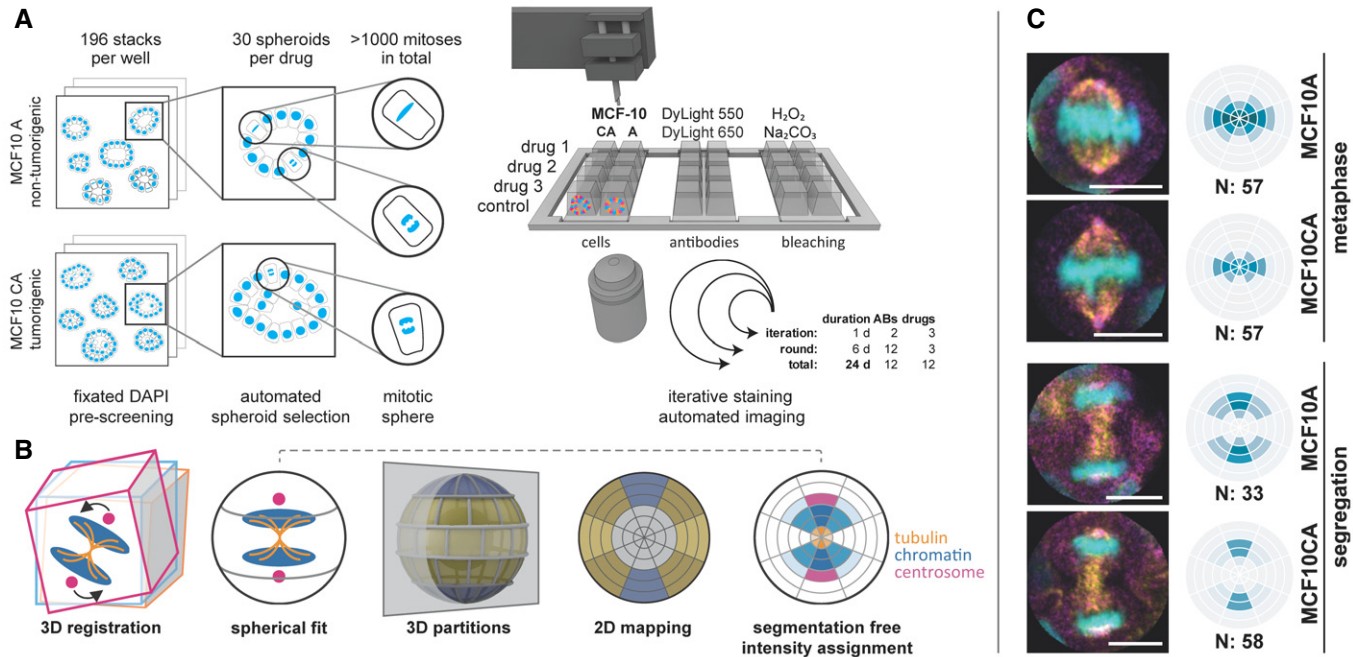

**Figure 1. Workflow of iterative antibody labeling.**

A After 48 h of drug treatment, MCF10A and MCF10CA cells were fixed and DAPI stained. Pre-screening comprised 196 image stacks per well to automatically select 30 spheroids that each showed at least one mitosis. At each round, selected positions for three drug treatments plus control were stained, imaged, and bleached in six iterations with two antibodies each. Within 24 days, we acquired 3D stacks of 12 antibodies on 12 drug treatments and two cell lines. ABs, antibodies.

B SpheriCell visualization: Stacks were 3D registered and a sphere was fitted to each cell division area, which was partitioned into three symmetrical sets of spherical sectors (equatorial, diagonal, polar) and six equidistant shells. Spherical 3D localization can be visualized by a longitudinal cut resulting in a 2D polar grid that contains projected mean values of 3D partitions. Moreover, cell poles are not distinguishable, so the results are centrically symmetric. Localization of mitotic proteins can be intuitively determined from the 2D projected partitions as exemplified by tubulin, chromatin, and centrosomal regions. Color intensities reflect normalized, mean protein concentrations in each bin.

C Example images and DAPI binning. Distinguished between MCF10A and MCF10CA, and metaphase and segregation spanning ana- and telophase. DAPI (cyan), γ-tubulin (magenta), and β-tubulin (yellow). For visualization, images were rotated to vertically align mitotic axes. *N*, number of mitoses contributing to mean values (scale bars: 10 μm).

## Studying effects of chemotherapeutic drugs on intracellular spatial distributions of proteins

Next, we applied the 3D SPECS approach for quantitatively studying effects of cancer drugs on the topography of mitotic phases. We analyzed the effects of twelve targeted inhibitors on mitotic kinases or effector proteins of dividing MCF10A and MCF10CA cells (Fig 3), specifically on protein concentrations and preferred localizations. To compare spatial distribution patterns of protein intensities, in each cell, 18 subcellular spherical ROIs were defined by a combination of six eccentricity shells with three orientations relative to the division plane, scaled per cell to a sphere volume. In analogy to calculating a center of mass, we specified measures of spatial intensity distributions (see Materials and Methods). We visualized significant concentration fold changes and spatial changes in eccentricity and orientation compared between cell lines, mitotic phases, and for inhibitors relative to controls (Fig 3A and B). All visualized measures, 95% confidence intervals, *P*-values, and numbers of mitotic events are available in Dataset EV1. Strikingly, changes in eccentricity and orientation of the localization pattern could be observed in MCF10CA relative to MCF10A cells mostly during metaphase, whereas almost no differences in spatial distributions were observed during segregation (Fig 3A, MCF10CA vs. MCF10A). Obviously, as indicated for the comparison between segregation and metaphase, the mitotic phase strongly influenced the spatial distributions of most observed proteins (Fig 3A, seg. vs. meta). Changes in the distribution pattern for DAPI and γ-H2AX reflect the movement of the nucleus toward the cell division axis and to higher eccentricity, while the other proteins move closer to the cell division plane. During segregation, relative to metaphase, both cell lines showed an expected decrease in CDC20 concentration (Sullivan & Morgan, 2007) and elevated BIRC5 concentrations, whereas CENP-A was only increased in MCF10A cells.

Notably, inhibitor treatments resulted in more pronounced effects on concentration fold changes (Fig 3B) than on spatial distributions (Fig EV2), only Haspin, Aurora B (eccentricity), and PLK1 (orientation) being notable exceptions. For these inhibitors, exemplary SpheriCell maps visualizing significant fold changes in ROIs are shown in Fig 3C, while SpheriCell maps of effects for all inhibitors and all measured species are presented in Figs EV3 and EV4. To facilitate comparisons, Fig 3D summarizes significant concentration fold changes, visualized in Fig 3B, that were exclusively

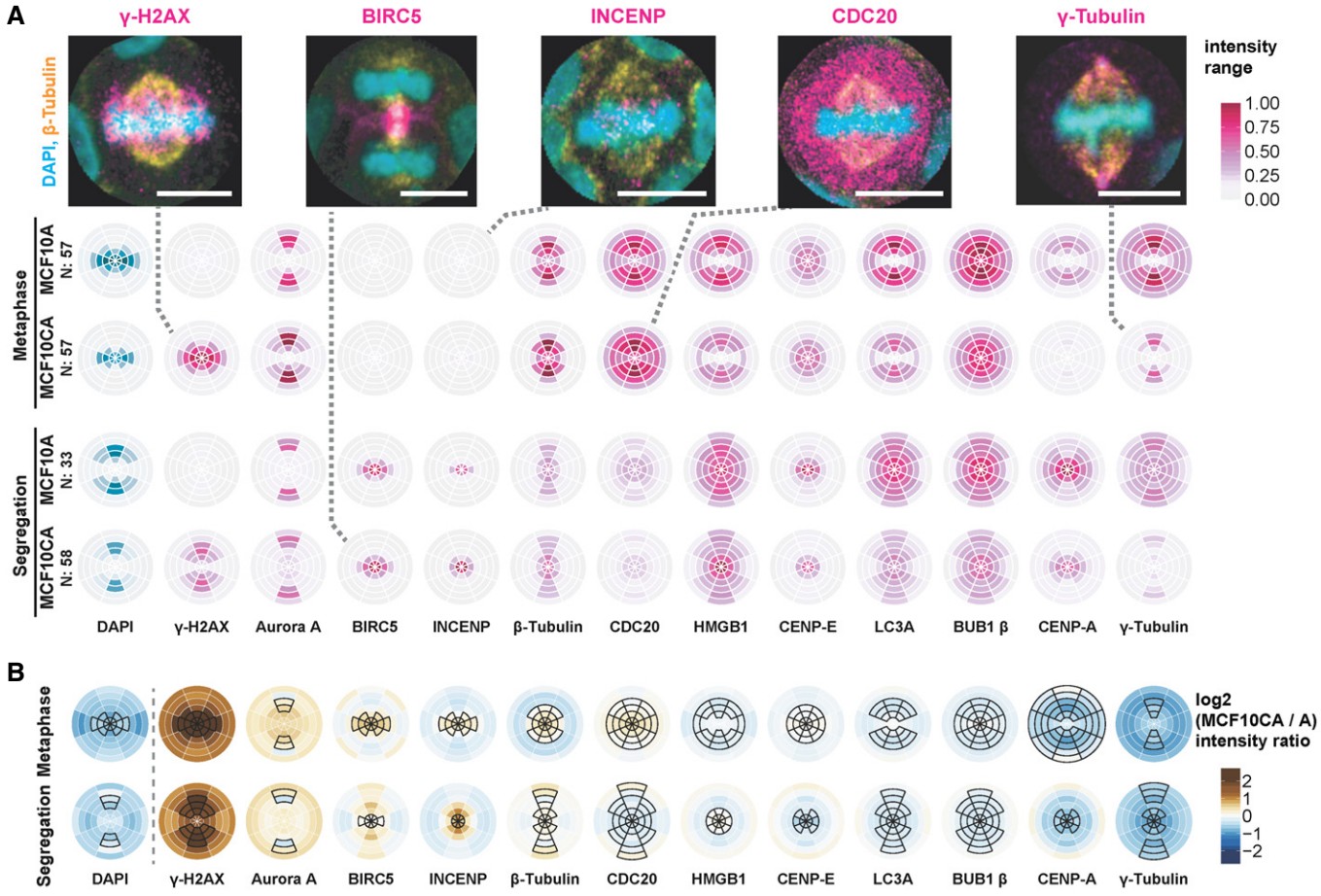

**Figure 2.  Localization and intensity changes in untreated MCF10A and MCF10CA cells.**

A    Localization of epitopes of twelve antibodies, besides DAPI staining, during metaphase and segregation (comprising ana- and telophase). SpheriCell plots depict mean intensity values across all imaging rounds. Stainings of proteins were ordered by decreasing difference between MCF10CA and MCF10A cells. Intensity ranges were specific to the antibody and are shown normalized between 0 and 1, effectively across all values of a column in the figure. Distribution patterns generally reflect the localization of individual proteins described before. Dashed lines connect SpheriCell plots with example images of antibody stainings (magenta), DAPI (cyan), and β-tubulin (yellow) (scale bars: 10 μm). LC3A: microtubule-associated proteins 1A/1B light chain 3A.

B    MCF10CA shows altered intensity patterns compared to MCF10A. SpheriCell plots depict differences of log2 transformed fluorescence intensity of MCF10CA and MCF10A [log2(CA) − log2(A)] for metaphase and segregation, in decreasing order. Black framed partitions indicate intensity distributions in untreated control images. LC3A, microtubule-associated proteins 1A/1B light chain 3A.

observed in MCF10A or MCF10CA cells, or in both cell lines. Here, the inhibition of master regulator Aurora kinase B, and also inhibition of Haspin known to be implicated in Aurora B positioning (Carmena *et al*, 2012), showed a prominent effect across nearly all proteins in MCF10A as well as MCF10CA cells. Moreover, we detected broad effects of increased DNA damage by Topoisomerase II poisoning (Nitiss, 2009a). Interestingly, MCF10CA cells appeared to be more sensitive to mitotic spindle interference, reflected by effects on inhibitors of Aurora A or PLK1, and the microtubule inhibitor paclitaxel (Fig 3D). Furthermore, inhibition of CHK1 affected only proteins in MCF10CA spheroids. Contrarily, although KIF11 (Eg5) and KIFC1 (HSET) facilitate separation and clustering of centrosomes (Rath & Kozielski, 2012), the effects due to inhibition of KIF11 were restricted to MCF10A cells. High natural levels of γ-H2AX intensity in MCF10CA were not increased by treatments as observed in MCF10A cells, and similarly, BIRC5 concentration was

only affected in MCF10A. Analogous evaluations were conducted with regard to measures of abundances, obtained by weighting ROI intensities according to their volumes (Appendix Fig S1, see Materials and Methods). Only one inhibitor, Haspin, had slight effects on cell volume estimates. For all other inhibitors, observations with regard to fold changes of abundances or concentrations were similar.

We conclude that applying the established 3D SPECs approach on mitotic cells resulted in a detailed and differentiated picture of effects dependent on cell lines, mitotic phases, and inhibitor effects.

### Modeling intracellular distribution maps of proteins involved in mitosis

To gain a mechanistic explanation for the measured intracellular distributions, we developed a non-linear model that was calibrated

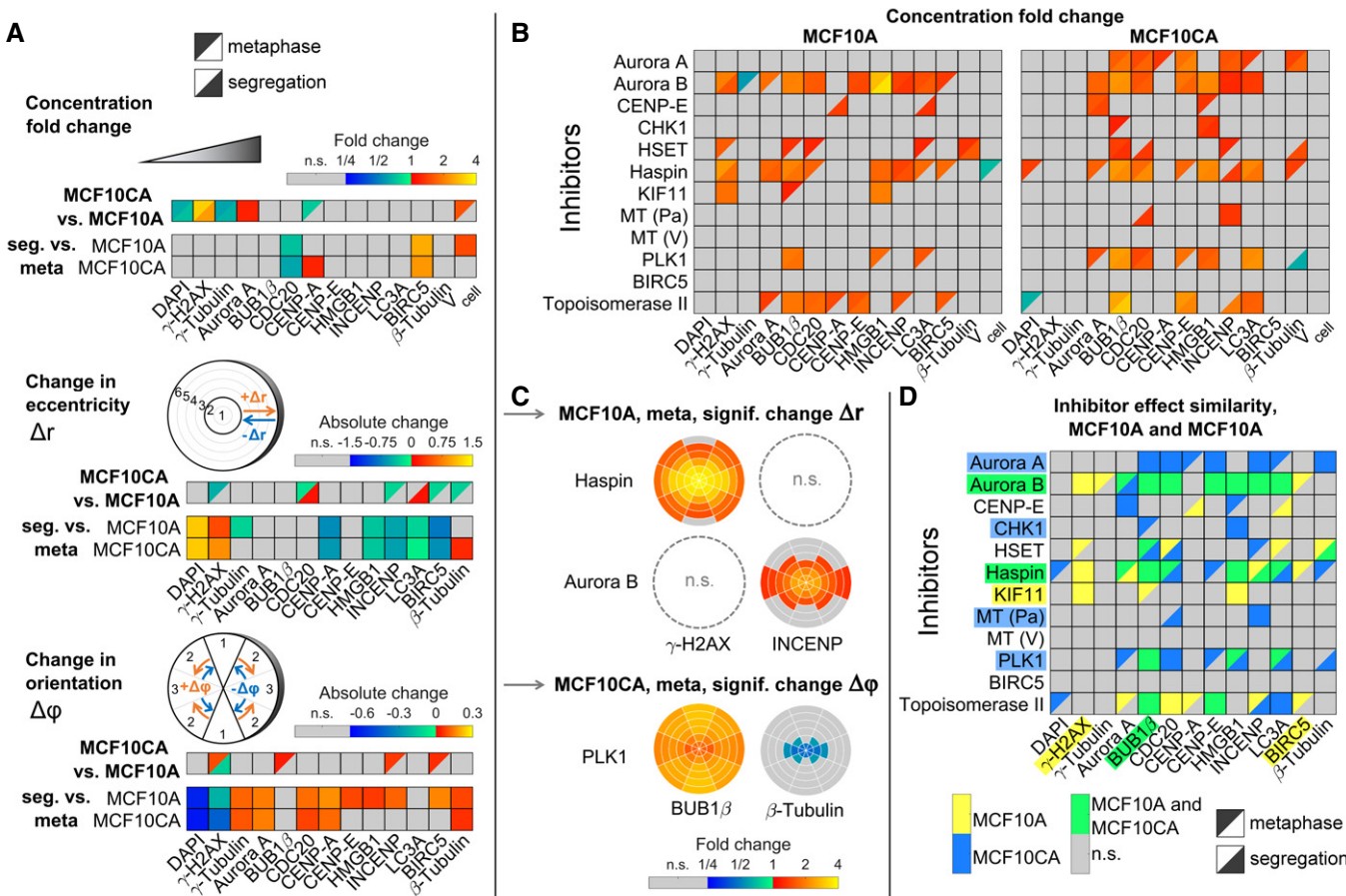

**Figure 3. Topographical effects of cell line, mitotic phase, and inhibitors.**

A Concentration fold changes and localization changes, quantified as changes in eccentricity and orientation of localization patterns for comparisons between cell lines (MCF10CA vs. MCF10A), mitotic phases (segregation vs. metaphase). Effects related to cell line were indicated separately for metaphase (upper left triangles) and segregation (lower right triangles). The upper panel shows color-coded fold changes in average concentrations (total intensities normalized by cell volumes) for DAPI and antibody stainings, together with fold changes of the cell volume, on a logarithmic scale. In the panel below, eccentricity changes for intensity distributions in spherical ROIs were visualized. Positive values describe a movement to the periphery, while negative values represent a movement to the center of the cell. Similarly, in the bottom panel, changes in angular orientation of intensity distributions were visualized. Positive values describe a movement toward the plane perpendicular to the cell division axis, while negative values describe a movement toward the cell division axis. In cases of significant differences to negative controls (Welch's *t*-test with $P < 0.05$, Bonferroni multiple testing correction for 52 comparisons in each measured species), fold changes relative to negative controls are indicated by colors (n.s., not significant; seg., segregation).

B Fold changes for inhibitor treatments (inhibitor vs. control) for MCF10A (left) and MCF10CA cells (right) as in (A). Analogous measures in eccentricity and orientation changes are shown in Fig EV2 (MT, microtubule inhibitor; Pa, paclitaxel; V, vinblastine).

C Exemplary SpheriCell plots showing fold changes in ROIs for inhibitor treatments relative to controls. A significant decrease in the eccentricity of distribution patterns $\Delta r$, due to pronounced concentration increase in central ROIs, was observed for γ-H2AX upon Haspin inhibitor treatment and for INCENP upon Aurora B inhibitor treatment (n.s., no significant change in $\Delta r$). A significant decrease in the measure of the distribution pattern orientation $\Delta\phi$, equivalent to an arrangement toward the cell division axis, was observed for BUB1β and β-tubulin upon treatment with PLK1 inhibitor. All SpheriCell plots for fold changes in response to inhibitor treatments are shown in Figs EV3 and EV4.

D Overlay of significant inhibitor effects in MCF10A cells, MCF10CA cells, or MCF10A and MCF10CA cells. Color highlighted proteins denote predominant effects per row (inhibitors) or column (measured species).

with our dataset of spatially resolved fluorescence intensity measurements of proteins involved in mitosis in combination with DAPI fluorescence. The model describes concentrations for monomers, homo-, and heterodimers of all measured species in spherical ROIs, defined by SpheriCell maps. In the following, we will give an overview about the model implementation and calibration (see Materials and Methods for details). The model explains the recruitment of $i = 1\ldots13$ measured species to $j = 1\ldots18$ mitotic ROIs by

first-order reactions with affinity parameters $\alpha_{il}$. In mitotic ROIs, all stained species can form homo- or heterodimers, described by second-order reactions with affinity parameters $\beta_{ij}$ (Fig 4A). Affinity parameters were defined as the inverse of dissociation constants for recruitment to mitotic ROIs or for dimerization reactions. Of note, affinities between species were taken only into account for explaining the local enrichment of proteins but do not necessarily imply biochemical interactions between proteins. Reactions were assumed

in steady state in agreement with the observation that diffusion, association, and dissociation reactions of the measured species are typically fast compared to the timescale of biochemical reactions involved in mitosis (Wachsmuth *et al*, 2015). At first, affinities of every protein to mitotic ROIs and affinities between proteins were estimated by model fitting to fluorescence intensity measurements in spherical ROIs of untreated cells.

To predict new affinities between proteins, we fitted a model of interactions from literature in *Ingenuity* Pathway Analysis (IPA; Krämer *et al*, 2014) regarded as ground truth. This initial model was fitted to our untreated control cells. Then, by sequential forward selection, new mutual affinities between proteins were additionally included in the model and pertained if model fits were significantly improved, based on likelihood-ratio testing. Fig 4B visualizes the 19 known interactions overlaid with all 16 additionally predicted mutual affinities. For example, we identified known association of γ-tubulin with CDC20 (Müller *et al*, 2006) as well as known DNA-binding of BIRC5 (Carmena *et al*, 2012). While we triggered DNA damage pathways with Topoisomerase II poisoning and inhibition of CHK1 (Nitiss, 2009b), activation of DNA repair mechanisms could be inferred from double-strand break marker γ-H2AX (Paull *et al*, 2000), necrosis-associated HMGB1 (Pallier *et al*, 2003), and autophagic vesicle marker MAP1LC3A (LC3A). BIRC5 was predicted by the model to interact with LC3A, which links mitotic surveillance and autophagy pathways (Mariño *et al*, 2014). The model predicted interactions of γ-H2AX, γ-tubulin, and β-tubulin with several other proteins, which might indicate indirect interactions with the mitotic spindle or the cytoskeleton. Further coefficients describing mutual affinities between proteins are shown in Fig 4C, and estimated affinities to mitotic ROIs in Appendix Fig S2. Importantly, the fraction of a protein that is localized to a mitotic bin due to mutual interactions with other proteins does not only depend on affinity coefficients, but may represent high affinities of interacting species to the respective mitotic bin. The highest values of mutual affinities with other proteins were found between CENP-E molecules as described earlier (Chan *et al*, 1998) and for the newly predicted binding of β-tubulin to HMGB1. We further tested, which affinities according to reported interactions between species significantly contributed to explaining the experimental dataset. To this end, affinity parameters were withdrawn and the model was refitted to determine the difference in $\chi^2$ (Appendix Fig S3). Thereby, we found that only four affinities according to literature interactions significantly contributed to explaining the measured intensity distributions (marked by squares in Fig 4B). Notably, if an affinity parameter did not contribute to explaining the dataset, this does not imply the absence of binding between these species but is likely due to non-identifiability. Erroneously rejecting an affinity parameter that might have been determined by other experimental techniques rather results from insufficient discrimination between linear and second-order terms when fitting to measurements in single-cell ROIs.

We next inspected changes in affinity parameter estimates between mitotic proteins and their affinities to mitotic ROIs upon drug treatment. To this end, the model with known and additionally predicted interactions was fitted to datasets from cells treated with inhibitors to estimate mutual affinities between proteins and to mitotic ROIs. We observed that inhibitors generally affected mutual affinity coefficients indicated by differences between coefficient

estimates for untreated cells and average estimates for inhibitor treatments (Fig 4C and Appendix Fig S4). In comparison with effects from other inhibitor treatments, inhibition of PLK1 affected the localization patterns of several proteins and caused a strong specific shift in mutual affinities among several studied proteins in comparison with untreated cells (Fig 4D). Contrarily, affinities to mitotic ROIs during metaphase and segregation showed almost no differences to untreated cells (Appendix Fig S2E–H). It is tempting to speculate that effects of PLK1 inhibition are mediated through its involvement in spindle network formation (Zitouni *et al*, 2014). Specifically, predicted affinity of β-tubulin to γ-H2AX, HMGB1, and INCENP decreased, and chromosome affinity of BIRC5 appears to be reduced by inhibition of PLK1, whereas the predicted affinity of INCENP to HMGB1 is increased (Fig 4D). Reduced chromosome affinity of BIRC5 after PLK1 inhibition is in accordance with the finding that phosphorylation of BIRC5 by PLK1 is required for a proper chromosome alignment during mitosis (Carmena *et al*, 2012).

Taken together, we established a simple mathematical model that describes the recruitment of stained species to spherical ROIs as well as homo- and heterodimerization between species to explain observed intracellular distributions of proteins involved in mitosis. Sequential model extension could be used to extend a set of known interactions. Predictions of mutual affinities between the observed proteins involved in mitosis can be used to guide further experiments for investigating functional relations and protein complexes that are linked to cellular processes.

## Discussion

We developed 3D SPECS, a high-content screening assay employing automated iterative antibody labeling in 3D cell cultures. It allowed us to compare system-wide interactions between 12 proteins of two cell lines in two mitotic phases, upon 12 individual treatments. High automation comprises detection of mitoses, iterative staining and imaging, 3D partitioning, modeling, and visualization using SpheriCell, a novel approach that does not require image segmentation. Morphometric image processing operations as elastic registration are not necessary because spherical ROIs are defined individually for each cell based on automatically detected mitotic axes and spherical fits. This explorative approach recapitulated *prior* knowledge on proteins involved in mitosis and allowed the generation of novel hypotheses in mitotic pathway signaling.

Most prominently, we discovered upregulation of γ-H2AX in tumorigenic MCF10CA cells compared to MCF10A. Further, γ-H2AX was stronger affected by inhibitor treatments in MCF10A, which in turn appears to have a more robust spindle apparatus. Our novel combined imaging and mathematical modeling approach allowed us to disentangle inhibitor-mediated protein localization and binding affinity changes. It showed that changes in affinities between proteins due to inhibitor treatments were more pronounced than changes in individual protein localizations (Appendix Figs S2E–H), which can be interpreted as robustness of the architecture of cellular processes. In one specific example, we focused on the measured inhibitions of PLK1 activity, responsible for establishing the mitotic spindle and that is frequently hyper-activated in cancer (Kumar *et al*, 2017). Subsequent reduction in chromatin affinity of BIRC5

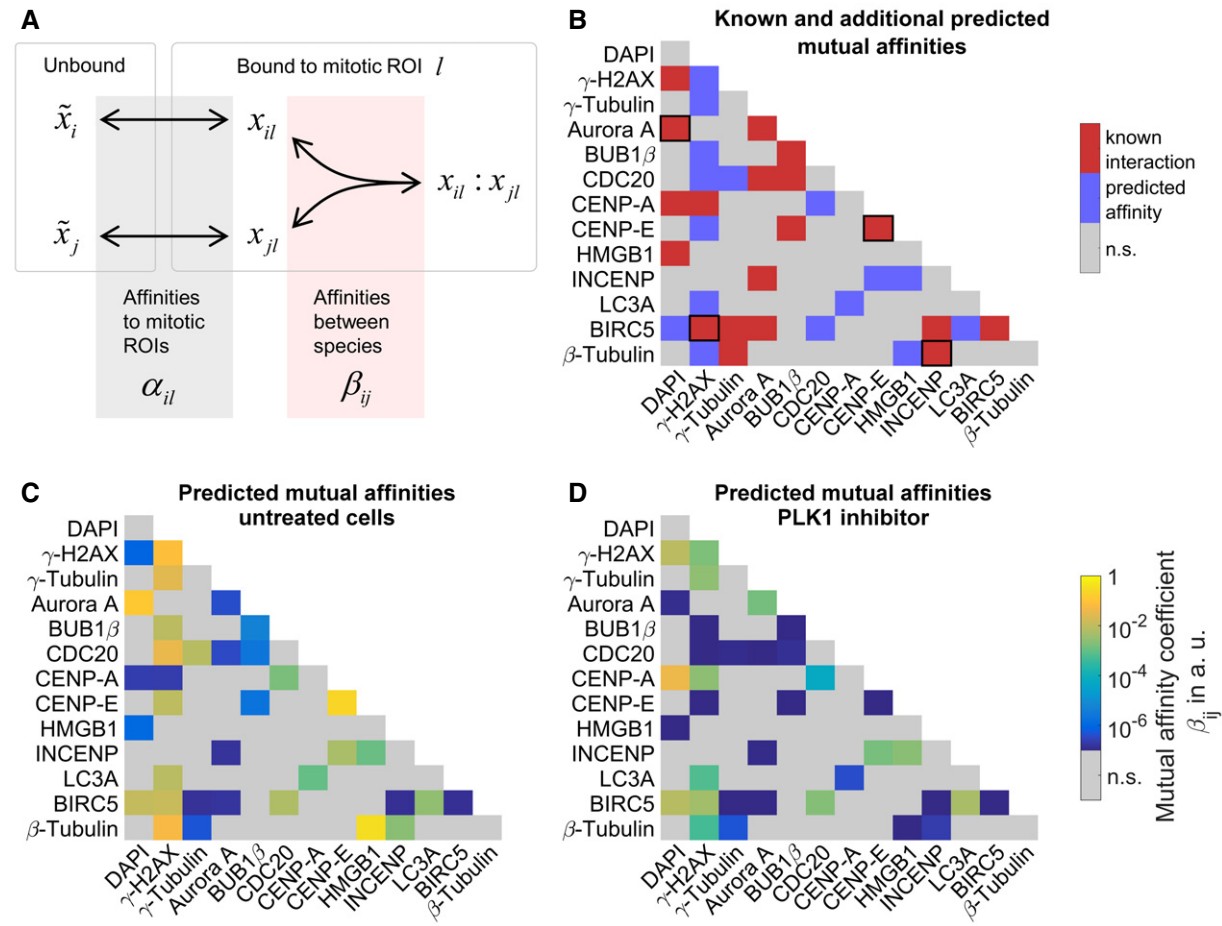

**Figure 4. Mathematical modeling of affinities between measured species.**

A Schematic graph of the mathematical model describing concentration distributions of measured species in mitotic ROIs. Spatial distributions are explained by affinities of species to cellular structures contained in mitotic ROIs $\alpha_{il}$ as well as homo- or heterodimeric interactions in ROIs described by affinities $\beta_{ij}$. Affinities are defined as the inverse of dissociation constants ($\tilde{x}_i$, unbound species $i$; $x_{il}$, bound species $i$ in ROI $l$, $x_{il}$:$x_{jl}$, heterodimer of species $i$ and $j$ in ROI $l$; see Materials and Methods for details).

B Affinities related to known protein–protein interactions from *Ingenuity Pathway Analysis* overlaid with additional predicted mutual affinities between measured proteins. Known affinities that significantly contributed to explaining the measured intensity distributions were marked by black squares. For affinities to mitotic ROIs, see Appendix Fig S2.

C Estimates of mutual affinities between measured proteins for untreated cells.

D Estimated mutual affinities between measured proteins after treatment with PLK1 inhibitor.

could be explained by its dependency on PLK1 phosphorylation (Carmena *et al*, 2012), most likely intertwined with its CPC function.

In cases, in which associations of proteins were predicted, especially in those involving γ-H2AX, β-tubulin, or γ-tubulin, it is likely that these proteins were present in larger multi-protein complexes. In such cases, associations might be undetectable with established biophysical techniques as FRET or FCCS because distances between proteins will exceed the required proximity. In the future, such associations in larger protein complexes might be determined by *in vivo* super-resolution microscopy.

We did not analyze effects of inhibitors on fractions of cells in different mitotic phases since we did not select mitotic cells in a randomized manner. It would be, however, interesting to link effects of inhibitors on intracellular distributions of proteins

involved in mitosis with effects on the duration of mitotic phases. Moreover, it might be interesting to further study model refinements related to treatment groups or investigate patterns of effects from inhibitor treatments. Our method can be readily extended to determine the activity of proteins by phospho-specific antibodies. For a more fine-grained assessment of protein localization, additional nuclear or membrane labels can be integrated into 3D SPECS. The SpheriCell approach that delivers intuitively simple and comprehensive visualization of protein localization in cell division can also be amended by including cell polarity landmarks, e.g., Golgi apparatus or ciliation of non-dividing cells. Taken together, we have demonstrated 3D SPECS as a novel workflow unraveling thus unprecedented levels of details in changes of protein localization and interaction upon drug treatment of three-dimensional cell cultures.

# Materials and Methods

## Reagents and Tools table

| Reagent/Resource | Reference or Source | Identifier or Catalog Number |
|---|---|---|
| **Experimental Models** | | |
| MCF10A pBabePuro (*Homo sapiens*) | Zev Gartner Lab | n/a |
| MCF10CA1d.cl1 (*H. sapiens*) | Barbara Ann Karmanos Cancer Institute | n/a |
| **Antibodies** | | |
| CENP-E (1:400, mouse monoclonal, clone 1H12) | Abnova | MAB1924 |
| BubR1 (1:600, mouse monoclonal, clone 8G1) | Thermo Fisher | MA5-16036 |
| beta-Tubulin (1:5,000, mouse monoclonal, clone TUB 2.1) | Abcam | ab11309 |
| CDC20 (1:400, rabbit polyclonal) | Bethyl | A301-179A |
| gamma-Tubulin (1:12,000, rabbit polyclonal) | Abcam | ab176404 |
| LC3A (1:400, rabbit polyclonal) | Novus | NB100-2331 |
| Survivin (1:1,000, rabbit monoclonal, clone EP2880Y) | Abcam | ab176402 |
| INCENP (1:1,000, mouse monoclonal, clone 3D2) | Thermo Fisher | MA5-17100 |
| Aurora A (1:6,000, rabbit monoclonal, clone EPR5026) | Abcam | ab176375 |
| CENP-A (1:500, rabbit polyclonal) | Abnova | PAB18324 |
| HMGB1 (1:3,000, rabbit monoclonal, clone EPR3507) | Abcam | ab176398 |
| $\gamma$-H2AX (1:2,500, rabbit monoclonal, clone 20E3) | Cell Signaling | 9718BF |
| **Chemicals, enzymes and other reagents** | | |
| **Matrigel** | | |
| Matrigel, growth factor reduced, phenol red-free | Corning | 356231 |
| Antibody conjugation | | |
| DyLight 550 Microscale labeling kit | Thermo Fisher | 84531 |
| DyLight 650 Microscale labeling kit | Thermo Fisher | 84536 |
| **Medium** | | |
| DMEM/F12, no phenol red | Gibco | 21041-33 |
| Horse Serum | Gibco | 16050-122 |
| EGF | Sigma | E9644-.2MG |
| Hydrocortisone | Sigma | H0888-1g |
| Cholera Toxin | Sigma | C8052-1MG |
| Insulin | Life Technologies | 12585014 |
| **Drugs** | | |
| Barasertib (1.11 nM) | SelleckChem | S1147 |
| CHR-6494 (500 nM) | MedChem Express | HY-15217 |
| CW069 (25.0 µM) | SelleckChem | S7336 |
| Etoposide (333 nM) | SelleckChem | S1225 |
| GSK461364 (2.20 nM) | SelleckChem | S2193 |
| GSK923295 (3.20 nM) | SelleckChem | S7090 |
| Ispinesib (1.70 nM) | SelleckChem | S1452 |
| MK-5108 (0.576 nM) | SelleckChem | S2770 |
| MK-8776 (9.00 nM) | SelleckChem | S2735 |
| Paclitaxel (2.67 nM) | SelleckChem | S1150 |
| Vinblastine (2.40 nM) | Sigma | V1377 |
| YM155 (0.540 nM) | SelleckChem | S1130 |

**Reagents and Tools table**  (continued)

| Reagent/Resource | Reference or Source | Identifier or Catalog Number |
|---|---|---|
| **Software** | | |
| R | https://r-project.org | 3.3.3 |
| Knime | https://www.knime.com | 3.2.1 |
| Python | https://python.org | 3.4 |
| Zeiss Zen blue | http://zeiss.com | 2012 |
| Zeiss Zen black | http://zeiss.com | 2012 |
| LabView | http://ni.com | 2013 SP1 |
| Fiji/ImageJ | https://fiji.sc | fiji-latest, timestamp 20160902160412 |
| MATLAB | https://www.mathworks.com/ | 2015a |
| IPA | https://www.qiagenbioinformatics.com/products/ingenuity-pathway-analysis/ | |
| **Other** | | |
| Axio Observer.Z1 | Carl Zeiss Microscopy | 431007 9901 000 |
| Scan Module LSM 710 Base Unit | Carl Zeiss Microscopy | 000000 1410 052 |
| Spectral Detection Unit LSM 780 + 2 PMT | Carl Zeiss Microscopy | 000000 1670 619 |
| Objective Plan-Apochromat 20×/0.8 M27 | Carl Zeiss Microscopy | 420650 9901 000 |
| Laser diode 405 nm CW 30 mW | Carl Zeiss Microscopy | 000000 1410 119 |
| Laser 561 nm | Carl Zeiss Microscopy | 000000 1410 117 |
| Master Beam Splitter Wheel for Laser 633, 561, 514, 488, 458 nm | Carl Zeiss Microscopy | 000000 1583 982 |
| Beam path NLO+405/445 Axio Observer/Axio Imager | Carl Zeiss Microscopy | 000000 1741 731 |
| Spinning Disc Unit CSU-X1A 5000 | Carl Zeiss Microscopy | 423638 9090 000 |
| Spinning Disc Laser Unit 405 nm | Carl Zeiss Microscopy | 000000 1514 464 |
| Rolera EM-C2 Bio-Imaging Microscopy Camera, Mono, 14-bit | Carl Zeiss Microscopy | 000000 1915 668 |
| PROcellcare 5030 + Peristaltic Pump Extension Unit | PROdesign | n/a |
| Energenie EG-PM2 | Energenie | EG-PM2 |
| 4× LabTek holder | EMBLEM | LTT-01 and LTT-02 |

## Methods and Protocols

### Iterative staining and imaging

Mitotic proteins were assessed after 48-h drug treatment by iterative immunofluorescence labeling. The antibodies were either labeled with one of DyLight 550/Cy3 or DyLight 650/Cy5. Both types of dyes could be used interchangeably in terms of excitation and emission spectra.

While Matrigel is essential for acinar growth of spheroids (Debnath & Brugge, 2005), it also dissolves quickly when the bleaching solution is applied. Therefore, we have used DyLight instead of Cy or Alexa (Lin *et al*, 2015) labeled antibodies, as they bleach much faster and have a very strong fluorescence signal nevertheless. Applying the bleaching solution significantly longer than 5 min at a time typically dissolved the Matrigel carrying the spheroids.

All treatments were imaged at 30 spheroids that showed at least one mitosis each. A total of 196 stacks per well, eight wells per round, and four rounds resulted in 6,272 image stacks with 21 slices each that were automatically pre-screened for mitotic events. Iterative high-resolution images of 30 positions per well in eight wells in each of four rounds totaled in 960 identified spheroids that were imaged each with 31 slices, after staining and after bleaching in six iterations.

For analysis and visualization, every mitosis was aligned along its division plane for a spherical neighborhood that contains the cell division in equatorial axis (see Fig 1B).

### 3D cell culture and drug treatment

Human mammary epithelial MCF10A pBabePuro cells were kindly obtained from Zev Gartner Lab; MCF10CA1d.cl1 (MCF10CA) cells from Karmanos Cancer Institute. Eight well Lab-Tek Chambered Coverglass slides (Sigma 155411) were treated with 2 M NaOH for 20 min and rinsed twice for 10 min with MilliQ water. Ten µl Matrigel (growth factor reduced, phenol red-free, Corning 356231) per well was added on ice with pipette tips pre-cooled to −20°C. MCF10A and CA cells were seeded with 2% Matrigel in Growth Medium overnight. Growth Medium was

adapted from Debnath *et al* (2003) and is based on DMEM/F12 (no phenol red, Gibco 21041-33), with 5% Horse Serum (Gibco 16050-122), 20 ng/ml EGF (Sigma E9644-.2MG), 0.5 mg/ml Hydrocortisone (Sigma H0888-1g), 100 ng/ml Cholera Toxin (Sigma C8052-1MG), and 10 µg/ml Insulin (Life Technologies 12585014). For the inhibition experiments, the cells were treated for 48 h at 1 day after seeding.

### Inhibitors

Drugs, suppliers, and concentrations used were Barasertib (Aurora B inhibitor; alternative name AZD1152-HQPS; SelleckChem S1147; 1.11 nM); CHR-6494 (Haspin inhibitor; MedChem Express HY-15217; 500 nM); CW069 (HSET inhibitor; SelleckChem S7336; 25.0 µM); Etoposide (Topoisomerase II inhibitor; SelleckChem S1225; 333 nM); GSK461364 (PLK1 inhibitor; SelleckChem S2193; 2.20 nM); GSK923295 (CENP-E inhibitor; SelleckChem S7090; 3.20 nM); Ispinesib (KIF11 inhibitor; alternative name SB-715992; SelleckChem S1452; 1.70 nM); MK-5108 (Aurora A inhibitor; alternative name VX-689; SelleckChem S2770; 0.576 nM); MK-8776 (CHK1 inhibitor; alternative name SCH 900776; SelleckChem S2735; 9.00 nM); Paclitaxel (microtubule inhibitor; SelleckChem S1150; 2.67 nM); Vinblastine (microtubule inhibitor; Sigma V1377; 2.40 nM); and YM155 (BIRC5 inhibitor; SelleckChem S1130; 0.540 nM).

### Antibodies and labeling kits

Antibodies were conjugated with DyLight 550 and 650 Microscale labeling kits per supplier reference manual (Sigma, 84531 and 84536, respectively) unless otherwise stated. Antibody targets, dilutions, supplier, and conjugation method in iterative staining order were CENP-E (1:400; Abnova MAB1924; conjugated DyLight 550); BubR1 (1:600; Thermo Fisher MA5-16036; pre-conjugated with DyLight 650); beta-tubulin (1:5,000; Abcam ab11309; pre-conjugated with Cy3); CDC20 (1:400; Bethyl A301-179A; conjugated DyLight 550); gamma-tubulin (1:12,000; Abcam ab176404; pre-conjugated with Cy3); LC3A, microtubule-associated proteins 1A/1B light chain 3A (1:400; Novus NB100-2331; conjugated DyLight 650); BIRC5 (1:1,000; Abcam ab176402; pre-conjugated with Cy3) INCENP (1:1,000; Thermo Fisher MA5-17100; conjugated DyLight 650); Aurora A (1:6,000; Abcam ab176375; pre-conjugated with Cy3); CENP-A (1:500; Abnova PAB18324; conjugated DyLight 650); HMGB1 (1:3,000; Abcam ab176398; pre-conjugated with Cy3); H2AX (1:2,500; Cell Signaling 9718BF; conjugated DyLight 650).

### Iterative antibody labeling

Cell fixation was based on a protocol from Debnath *et al* (2003), with 1.85% formaldehyde solution (Sigma 252549) added to the medium for 10 min. Cells were rinsed twice with PBS and permeabilized for 10 min at RT with 0.5% TX-100 pre-chilled to 8°C, washed three times with PBS-glycine (130 mM NaCl, 7 mM $Na_2HPO_4$, 3.5 mM $NaH_2PO_4$, 100 mM glycine) for 10 min, and blocked overnight at RT in a blocking solution consisting of IF-wash solution (Muthuswamy *et al*, 2001; 130 mM NaCl, 7 mM $Na_2HPO_4$, 3.5 mM $NaH_2PO_4$, 7.7 mM $NaN_3$, 0.1% BSA, 0.2% Triton X-100, 0.05% Tween-20) with 10% goat serum (Sigma G9023-10ML) and 1:1,000 DAPI (Sigma D8417-1MG), inside an opaque EMBL microscope incubation chamber. For each iteration, two antibodies were diluted in freshly prepared blocking solution and stored in a slide

within a 4× Lab-Tek holder (EMBLEM LTT-01 and LTT-02). They were automatically pipetted into the wells by a peristaltic pump of the ProCellcare 5030 system (ProDesign) and incubated for 3 h, washed twice with IF-wash for 5 min and three times with PBS for 5 min. After imaging, freshly prepared $H_2O_2$ bleaching solution (Gerdes *et al*, 2010) containing 3% $H_2O_2$ (AppliChem, Cat. No. 121076) and 0.1 M $Na_2CO_3/NaHCO_3$ buffer at pH ≈ 10 was stored in another Lab-Tek. It was automatically applied for 5 min and washed twice with PBS for 5 min. Standard incubator light source was switched on during bleaching with Energenie EG-PM2. Pipetting positions were planned with Zeiss Zen blue (www.zeiss.com/zen), and pipetting workflow was implemented in LabView (www.ni.com/labview).

### Pre-screen

During blocking, slides were imaged with a Yokogawa CSU-X1 spinning disk unit attached to a Zeiss Observer Z1 inside an EMBL incubation chamber; 196 image stacks of 401.6 × 400 × 60 µm were taken per well with a plan-apochromat 20×/0.8 NA objective. Stack slices had 1,004 × 1,002 pixels, step size was 3 µm, and exposure 40 ms. Candidate mitotic positions were detected via their DAPI signal by a custom KNIME workflow and selected or expanded manually if necessary. The automatic selection excludes monolayer slices and uses a supervised tree ensemble classifier (comparable to a random forest). For each treatment and cell line, 30 positions of spheroids with at least one mitosis each were selected for imaging during the iterative staining workflow.

### Acquisition of iterative staining images

After each round of bleaching or staining, spheroids were automatically imaged with a laser scanning confocal Zeiss LSM 780 connected to the same Axio Observer as the spinning disk unit. Stack dimensions were 106.07 × 106.07 × 60 µm, with 512 × 512 pixels per slice and 2 µm Z steps. Objective was plan-apochromat 20×/0.8 NA, pixel dwell time 3.15 µs, and pinhole 32 µm. Emission spectra were taken at 410–489 nm (DAPI), 560–586 nm, 586–612 nm, and 612–630 nm (three parts of Cy3/DyLight 550), and 638–758 nm (Cy5/DyLight 650). Visible light beam splitter was MBS 488/561/633, and invisible light beam splitter MBS 405.

### Image processing

Splitting the emission spectrum from 560 to 630 nm in three parts allowed for post-acquisition exposure correction. Only for BIRC5, it was necessary to exclude the strongest emission channel from the labeling image. All remaining split channels were averaged. All image processing steps were embedded in KNIME workflows (Berthold *et al*, 2008). To detect positions of cells, an Otsu threshold segmentation method was used to detect nucleus regions in the DAPI signal. The KNIME nodes `Global Thresholder` and `Image Segment Features` were used to extract all available image features. Based on these features, nucleus regions were classified as mitoses by applying a machine learning algorithm. For this purpose, the KNIME node `Tree Ensemble Predictor` was used after training the node `Tree Ensemble Learner` on a ground truth dataset of manually assigned mitoses. Thereby, mitoses were detected in 196 stacks per inhibitor with 60 slides per stack. The next steps are provided as protocol to apply the image processing workflows

provided in Dataset EV2 on a sample dataset available in the BioStudies database (https://www.ebi.ac.uk/biostudies/) with identifier S-BSST176.

- Segment DAPI signal in 3D with `1 - 3d segmentation overview + storage` workflow. It employs a region growing algorithm (Berthold *et al*, 2008).
  - Manually set seeds within mitotic positions either in 2D projections or in case of overlaps, in 3D images.
  - Add borders to closely neighboring nuclei, especially in *z*-direction.
  - Annotate segmented areas with their mitotic phase. The workflow then joins ana-/telophases to collect cells in segregation.
  - Verify assignment of segregating split chromatin regions to a single dividing cell was verified with β-tubulin staining.
- Register consecutive stacks per imaging position with `2 - 3dRegWithCleanup`.
  - This workflow provides a batch compatible subpixel alignment using Fiji (Schindelin *et al*, 2012), its plugin `Correct 3D Drift` (Parslow *et al*, 2014) including multi time scale, subpixel, and edge enhancements, and plugin `MultiStackReg` (Thevenaz *et al*, 1998) with scaled rotation.
- Verify registration and annotate mitoses. If you want to create your own annotations, follow workflows `3a` through `3d`. Otherwise, proceed with workflow `3d` to use our annotations.
  - Exclude multiple mitoses per sphere ("overfilling"), and indistinct mitotic phases ("unclear"). Very early anaphases that started to segregate very recently are also to be skipped, as they would require an own class between metaphase and segregation.
  - `3a - registration tester` uses a custom virtual autofocus on DAPI that selects physically highest local maximum of variance. It antibody channels and allows to check for β-tubulin staining strength. Exclude missing cells due to loss of Matrigel or failed registration from further analysis.
  - Refine the selection with `3b - verify registration`. It shows the central slice of the image stacks and slices 40 and 20 interpolated steps above and below.
  - Annotate indistinct mitotic phases based on their DAPI signal and antibodies against Aurora A and β-tubulin using `3c - annotate mito areas`. Assign to metaphase or segregation if two centrosomes can be detected which are connected via β-tubulin to the chromatin regions.
  - Collect all single annotations of registration, staining strength, and telophase pairing with `3d - combine annotations`.
- Calculate SpheriCell partitioning with workflow `4 - sphericalAnalysis`.
  - In preparation of this workflow, generate spherical segment angles with Recursive Zonal Equal Area Sphere Partitioning Toolbox (EQSP; Leopardi, 2006) for 180 areas, or use our precomputed file `bins_180.oct`.
  - For the spherical neighborhood, the main workflow interpolates images linearly in Z to match the X/Y pixel dimensions.
  - Orientation of mitoses is identified with 3D ellipsoid fits. To this end, the KNIME workflow calls `3d_ellipsoid_fitting.ijm`, a batch compatible wrapper for calculation of ellipsoid fits using the 3D ImageJ suite (Ollion *et al*, 2013).
  - A custom embedded R script joins the EQSP areas to segments and to fits them in size and orientation to the individual mitoses. Metaphases could use those values as is, but the size of

segregating cells is overestimated by the ellipsoid fit and replaced by the centroid distances of their individual chromatin regions. Their 3D orientation uses the average of the first two eigenvectors and the normalized centroid to centroid vector as third. Subsequently, 3D segments were binned in three spherical intervals (equatorial [−30°, 30°] and [150°, 210°]; diagonal [30°, 60°], [120°, 150°], [210°, 240°], and [300°, 330°]; polar [60°, 120°] and [240°, 300°]; see Fig 1B). The six spherical neighborhood shells grow linearly in their radius from the mitotic center, and the inner four span the identified nucleus area.
- Combine 3D bin intensities and annotations to an R (R Core Team, 2017) representation using `multiSphere.R`, which employs R packages `data.table` (Dowle & Srinivasan, 2016), `plyr` (Wickham, 2011), and `stringr` (Wickham, 2017).
- Generate SpheriCell plots for drug and antibody effects with `makePlots.R`. This script makes use of R packages `ggplot2` (Wickham, 2009), `RColorBrewer` (Neuwirth, 2014), and `Cairo` (Urbanek & Horner, 2015).
- Complete output image data with a total size of about 150GB can be inspected via `Shiny` (Chang *et al*, 2016) web application in `ui.R` that shows SpheriCell plots in a responsive web interface. It can be accessed at https://ibios.dkfz-heidelberg.de/iterstain.
  - Upon selection of a SpheriCell plot, microscopy images of corresponding treated and untreated cells are shown side-by-side, each with three layers around the central slice. Uses `RBioFormats` (Oles, 2017), `EBImage` (Pau *et al*, 2010), and `devtools` (Wickham & Chang, 2016).

### Visualization

Antibody intensities are depicted as color-coded mean values for SpheriCell ROIs (Fig 1B). To avoid an artificial increase in background signal of antibodies, DAPI intensities below a minimum threshold were excluded. Highlighting of partitions was determined by the control intensity over all rounds. Data were visualized with the packages ggplot2 (Wickham, 2009), EBImage (Pau *et al*, 2010), and shiny (shiny.rstudio.com) for R (www.r-project.org).

### Evaluations of fluorescence measurements in SpheriCell ROIs

In each SpheriCell ROI, measured fluorescence values of associated voxels were averaged. These average intensities were assumed to be proportional to protein concentrations in these ROIs in accordance with basic assumptions for quantitative immunohistochemistry and quantitative fluorescence microscopy (True, 1988; Waters, 2009). To eliminate influences resulting from differences of antibody affinities and dye coupling efficiencies, effects of mitotic phases, cell lines, and inhibitors were analyzed based on scale-free magnitudes of fold changes and measures of eccentricity and orientation of protein distribution.

We compared DAPI and antibody staining fluorescence intensities between cell lines, mitotic phases, and for inhibitor treatments relative to controls. To analyze effects of protein intensities between cell lines, mitotic phases, and inhibitor treatments, we defined measures proportional to concentrations or abundances. Furthermore, to compare spatial protein distributions, we defined characteristic measures of eccentricity and orientation. For each cell, 18

ROIs were defined as intersections between six eccentricity shells with indices $\mu = 1\ldots6$ and three orientations relative to the division plane (equatorial $[-30°, 30°]$; diagonal $[30°, 60°]$; polar $[60°, 120°]$) with indices $\nu = 1\ldots3$. Measures proportional to single-cell concentrations $c$ were defined by weighting fluorescence intensities $I_{\mu\nu}$ with mitotic ROI volumes $V_{\mu\nu}$

$$c = \frac{\sum_{\mu=1}^{6} \sum_{\nu=1}^{3} I_{\mu\nu} V_{\mu\nu}}{\sum_{\mu=1}^{6} \sum_{\nu=1}^{3} V_{\mu\nu}}. \tag{1}$$

Analogously, measures proportional to abundances $a$ of proteins were defined by

$$a = \sum_{\mu=1}^{6} \sum_{\nu=1}^{3} I_{\mu\nu} V_{\mu\nu}. \tag{2}$$

Moreover, we introduced a measure that describes the eccentricity of a protein distribution pattern. It was denoted as center of eccentricity $r$ and defined by a sum of eccentricities $r_\mu$ weighted by fluorescence intensities $I_{\mu\nu}$

$$r = \frac{\sum_{\mu=1}^{6} \sum_{\nu=1}^{3} I_{\mu\nu} r_\mu}{I_c}, \tag{3}$$

with the total sum of intensities $I_c = \sum_{\mu=1}^{6} \sum_{\nu=1}^{3} I_{\mu\nu}$. Similarly, the center of orientation $\varphi$ of a protein distribution pattern was defined by

$$\varphi = \frac{\sum_{\mu=1}^{6} \sum_{\nu=1}^{3} I_{\mu\nu} \varphi_\nu}{I_c}. \tag{4}$$

In Figs 3A and B, and EV4, significant changes of $r$ and $\varphi$ dependent on cell lines, mitosis phases, and inhibitor treatments were visualized.

Protein concentrations in cells are typically log-normally distributed. For this reason, we log-transformed measures before statistical testing (Zhou & Gao, 1997; Choi, 2016). Assuming samples with unequal variances, we performed Welch's *t*-tests to statistically test for differences between conditions. From applying (Bonferroni) correction for multiple testing for a total of 52 comparisons based on measurements for each stained species, significance was defined by $P < 0.05/52 \approx 9.62 \times 10^{-4}$. Confidence intervals for fold changes, eccentricity changes, and orientation changes were estimated by bootstrapping. We determined 95% confidence intervals from 1,000 bootstrap samples (see Dataset EV1 including data shown in Figs 3A and B, and EV4).

### Mathematical model of protein affinities to mitotic ROIs and mutual affinities between proteins

To describe binding of proteins to mitotic ROIs and mutual binding of proteins within mitotic ROIs, we constructed a mathematical model derived from ordinary differential equations (ODEs). Here, we describe the concept and the implementation of this model.

In the simplest case, we consider binding of two proteins $A_1$ and $A_2$. Concentrations of free species are denoted by $A_1$ and $A_2$ (Appendix Fig S5). Binding to cellular structures contained in mitotic ROI $l$, results in $A_{11}$ and $A_{2l}$ with concentrations $A_{1l}$ and $A_{2l}$.

Kinetic parameters describing $A_1$ binding and unbinding to this compartment are $k_{1l}$ and $k_{-1l}$, while $A_2$ binding and unbinding is described by $k_{2l}$ and $k_{-2l}$. We assume that binding sites for proteins in a compartment are not limiting. The equation describing the concentration of free $A_1$ thus reads

$$\frac{dA_1}{dt} = -k_{1l} A_1 + k_{-1l} A_{1l}. \tag{5}$$

Dissociation constants $K_{1l} = k_{-1l}/k_{1l}$ and $K_{2l} = k_{-2l}/k_{2l}$ are thus dimensionless parameters. In steady state, the concentrations for $A_1$ and $A_2$ bound in mitotic ROI $l$ equal $A_{1l} = A_1/K_{1l}$ and $A_{2l} = A_2/K_{2l}$.

In mitotic ROI $l$, $A_{11}$ can reversibly bind to $A_{2l}$, resulting in $A_{1l}{:}A_{2l}$ with concentration $A_{1l}{:}A_{2l}$. We assume that binding between $A_{11}$ and $A_{2l}$ depends only on the interaction between the proteins but not on the mitotic ROI. Their binding and unbinding is described by parameters $\kappa_{12}$ and $\kappa_{-12}$, and the dissociation constant $\theta_{12} = \kappa_{-12}/\kappa_{12}$. ODEs for $A_{1l}$ and $A_{1l}{:}A_{2l}$ read

$$\frac{dA_{1l}}{dt} = k_{1l} A_1 - k_{-1l} A_{1l} - \kappa_{12} A_{1l} A_{2l} + \kappa_{-12} A_{1l}{:}A_{2l}, \tag{6}$$

$$\frac{dA_{1l}{:}A_{2l}}{dt} = \kappa_{12} A_{1l} A_{2l} - \kappa_{-12} A_{1l}{:}A_{2l}. \tag{7}$$

Solving at steady state for the concentration of $A_{11}{:}A_{2l}$ results in

$$A_{1l}{:}A_{2l} = \frac{A_1 A_2}{K_{1l} K_{2l} \theta_{12}}. \tag{8}$$

After immunofluorescence staining for $A_1$, the fluorescence intensity in mitotic ROI $l$ is therefore given by

$$I_{1l} = c_1 (A_{1l} + A_{1l}{:}A_{2l}) = c_1 \left( \frac{A_1}{K_{1l}} + \frac{A_1 A_2}{K_{1l} K_{2l} \theta_{12}} \right). \tag{9}$$

In this equation, the scaling factor $c_1$ relates concentrations to fluorescence intensity values. The measured intensity thus depends linearly on the concentration of $A_1$ and further contains the product of concentrations $A_1$ and $A_2$. This resembles that $A_1$ and $A_2$ are either recruited to mitotic ROI $l$ due to their affinity to this compartment or due to their mutual affinity.

To generalize this model, we describe binding of proteins $A_i$ with $i = 1\ldots n$ to mitotic ROIs $l = 1\ldots m$. Affinities of proteins $A_i$ to a mitotic ROI are described by the n-by-m matrix $\alpha_{il} = 1/K_{il}$, whereas affinities between proteins are given by the n-by-n matrix $\beta_{ij} = 1/\theta_{ij}$. Then, intensities of all species in all mitotic ROIs of a cell are given by

$$I_{il} = c_i \left( \frac{A_i}{K_{il}} + \sum_{j=1}^{n} \frac{A_i A_j}{K_{il} K_{jl} \theta_{ij}} \right) = c_i \left( \alpha_{il} A_i + \sum_{j=1}^{n} \alpha_{il} \alpha_{jl} \beta_{ij} A_i A_j \right). \tag{10}$$

For parameter estimations, we made the simplifying assumption that the concentrations of free proteins $A_i$ were approximately proportional to the average cellular concentrations $A_{i,t} = s_i A_i$, with the proportionality factor $s_i \geq 1$. This assumption holds true if the proteins are recruited to mitotic ROIs due to their direct affinities to the cellular structures contained in these ROIs rather than their affinities to the other observed proteins, which will be justified below.

Average cellular intensities were calculated by weighting intensities to mitotic ROIs $I_{il}$ with ROI volumes $V_l$, given by

$$I_{i,t} = \frac{\sum_{l=1}^{m} I_{il} V_l}{\sum_{l=1}^{m} V_l}. \tag{11}$$

For model fitting, we calculated fold changes relative to medians of average concentrations for the population of cells, $\hat{I}_{i,t}$, for intensities in mitotic ROIs, $\tilde{I}_{il} = I_{il}/\hat{I}_{i,t}$, and for average cellular concentrations, $\tilde{I}_{i,t} = I_{i,t}/\hat{I}_{i,t}$. Then, experimental measurements could be related to model variables $A_i$ by

$$\tilde{I}_{i,t} = \frac{I_{i,t}}{\hat{I}_{i,t}} = \frac{s_i c_i}{\hat{I}_{i,t}} A_i. \tag{12}$$

Thereby, equation (10) was rescaled to

$$\tilde{I}_{il} = \frac{c_i}{\hat{I}_{i,t}} \left( \frac{\alpha_{il} \hat{I}_{i,t}}{s_i c_i} \tilde{I}_{i,t} + \sum_{j=1}^{n} \frac{\alpha_{il} \hat{I}_{i,t}}{s_i c_i} \frac{\alpha_{jl} \hat{I}_{j,t}}{s_j c_j} \beta_{ij} \tilde{I}_{i,t} \tilde{I}_{j,t} \right) \equiv d_i \tilde{\alpha}_{il} \tilde{I}_{i,t} \left( 1 + \sum_{j=1}^{n} \tilde{\alpha}_{jl} \beta_{ij} \tilde{I}_{j,t} \right). \tag{13}$$

Therein, the rescaled parameter $d_i = c_i/\hat{I}_{i,t}$ was equal to the inverse of the median average cellular concentration, and $\tilde{\alpha}_{il} = \frac{\alpha_{il} \hat{I}_{i,t}}{s_i c_i}$ was equal to the median concentration of species $A_i$ that was bound in mitotic ROI $l$.

To account for different affinities of proteins to mitotic ROIs during metaphase and segregation, $\tilde{\alpha}_{meta,il}$ and $\tilde{\alpha}_{segr,il}$, as well as scaling factors $d_{meta,i}$ and $d_{segr,i}$, intensity measurements during these cell cycle phases were separately described by

$$\tilde{I}_{meta,il} = d_{meta,i} \tilde{\alpha}_{meta,il} \tilde{I}_{meta,i,t} \left( 1 + \sum_{j=1}^{n} \tilde{\alpha}_{meta,jl} \beta_{ij} \tilde{I}_{meta,j,t} \right), \tag{14}$$

and

$$\tilde{I}_{segr,il} = d_{segr,i} \tilde{\alpha}_{segr,il} \tilde{I}_{segr,i,t} \left( 1 + \sum_{j=1}^{n} \tilde{\alpha}_{segr,jl} \beta_{ij} \tilde{I}_{segr,j,t} \right). \tag{15}$$

Thereby, we assume that rescaled affinities to mitotic ROIs, but not mutual affinities between proteins $\beta_{ij}$, were dependent on the cell cycle phase. We simultaneously fitted equations (14) and (15) to experimental data for estimating $d_{meta,i}$, $d_{segr,i}$, $\tilde{\alpha}_{meta,il}$, $\tilde{\alpha}_{segr,il}$, and $\beta_{ij}$.

The interaction model was implemented in MATLAB (The Math-Works, Natick, MA, USA). For model calibration, we applied the solver lsqnonlin using the trust-region-reflective algorithm. To analyze relevant interactions between proteins, we first used experimental data from controls (cells not treated with inhibitors). Data from MCF10A and MCF10CA cells were fitted together, assuming that differences between cell lines were only dependent on different average cellular concentrations of all proteins but not on affinities to mitotic ROIs or affinities between proteins. A total of 513 to 529 parameters were estimated by model fitting to 50,635 data points (47,970 intensity measurements for mitotic ROIs and 2,665 average intensity values) from control experiments in 205 cells. To equally weight residuals for data points $\tilde{I}_{il}$ of different magnitudes, we

assumed the error model

$$\varepsilon(\tilde{I}_{il}) = 0.05 \cdot \tilde{I}_{il} + 0.05 \cdot \max(\tilde{I}_{il}), \tag{16}$$

assuming that for each measurement the experimental error is given by 5% of the measurement value plus 5% of the maximal value of all included cells. This procedure is commonly recommended if repeated measurements in the same objects as single cells are not available (Kreutz *et al*, 2007; Maiwald & Timmer, 2008). For model fitting, residuals between measurements and observables

$$r = \left( \frac{\tilde{I}_{il,data} - \tilde{I}_{il}}{\varepsilon(\tilde{I}_{il,data})} \right)^2, \tag{17}$$

for all cells were minimized. Initially, for all estimated parameters, large intervals between $10^{-7}$ and $10^2$ were allowed. Since none of the parameters $\tilde{\alpha}_{meta,il}$, $\tilde{\alpha}_{segr,il}$, $d_{meta,i}$, and $d_{segr,i}$ touched the lower interval boundary and all of these parameters were larger than $10^{-3}$, we restricted these parameters to the interval between $10^{-3}$ and $10^2$ to accelerate convergence in model fitting. Because estimates of several entries in $\beta_{ij}$ had smaller values, intervals between $10^{-7}$ and $10^2$ were pertained for these parameters. To accelerate convergence of model fits, parameters were fitted on a log-scale.

First, we started with fitting a model that only accounts for known literature interactions that were extracted from the *Ingenuity* pathway knowledge (IPA) database (Krämer *et al*, 2014). To this end, $\beta_{ij}$ was reduced to entries according to this set of literature interactions regarded as ground truth. Then, by sequential feature selection, additional affinities between proteins were further included if they could significantly improve the squared sum of residuals of the model fit. For each selection step of testing whether an additional interaction should be included, we performed 50 multi-start local optimizations by sampling initial conditions from allowed parameter intervals. We assured that after optimizations, differences between the best fits were below the residuals for single data points. Additional entries in $\beta_{ij}$ were selected based on likelihood-ratio testing, assuming that the likelihood-ratio for a model including an additional variable compared to a model without the additional parameter follows a one-dimensional $\chi^2$ distribution. An additional affinity between proteins was included in $\beta_{ij}$ if the increase in log-likelihood exceeded the 95% confidence interval of the cumulative one-dimensional $\chi^2$ distribution. Following this forward selection procedure, 16 additional affinities between proteins were included. The reduction of the residual sum of squares is shown in Appendix Fig S6.

These additional entries in $\beta_{ij}$ represent hypotheses about mutual binding between proteins. Notably, this predicted mutual binding may be distinct from possible functional interactions between proteins. Mutual affinity does not necessarily imply a functional interaction, whereas a functional interaction may not require high binding affinity. Nevertheless, predictions of mutual affinities between the observed proteins involved in mitosis can be used to guide further experiments for investigating functional relations and protein complexes that are linked to cellular processes.

After identifying an optimal set of additionally included affinities from model fitting to the control dataset, the model was fitted to data from inhibitor treatments. For every inhibitor treatment, the

parameters $d_{meta,i}$, $d_{segr,i}$, $\tilde{\alpha}_{meta,il}$, $\tilde{\alpha}_{segr,il}$, and the extended $\beta_{ij}$ were estimated by model fitting.

Finally, to estimate effects from inhibitor treatments on compartment affinities and on mutual affinities between proteins, we again fitted the model to the control dataset from untreated cells and to datasets from inhibitor treatments. We performed in each case 1,000 multi-start local optimizations by sampling initial conditions from allowed parameter intervals. Appendix Fig S7A shows the ordered sum of squared residual values for 1,000 multi-start local optimization runs for fitting the control dataset. In Appendix Fig S7B, model simulations for the best model fit to the control dataset were plotted against experimental data. It is evident that the model fit is highly consistent with experimental measurements. Known and additionally predicted entries of $\beta_{ij}$, and estimated affinity values for the control dataset were in Fig 4B and C. Estimated affinity values for an exemplary inhibitor (PLK1) were visualized in Fig 4D. Furthermore, the average affinities for all inhibitors were shown in Appendix Fig S2. Appendix Fig S2C shows estimates $\tilde{\alpha}_{meta,il}$, $\tilde{\alpha}_{segr,il}$ for estimated affinities to mitotic ROIs, whereas Appendix Fig S2D shows estimates $\alpha_{meta,il}/s_i$ and $\alpha_{segr,il}/s_i$ that were obtained by multiplying with scaling factors $d_{meta,i}$ and $d_{segr,i}$.

### Interpretation

Known interactions from literature were generated through the use of QIAGEN's Ingenuity® Pathway Analysis (IPA®, QIAGEN Redwood City, https://www.qiagenbioinformatics.com/products/ingenuity-pathway-analysis/). They are supported by at least one reference from the literature, from a textbook, or from canonical information stored in the Ingenuity Pathways Knowledge Base (Krämer *et al*, 2014).

## Data availability

The sample microscopy dataset from this publication has been deposited to the BioStudies database (https://www.ebi.ac.uk/biostudies/) and assigned the identifier S-BSST176. Custom software code for the 3D SPECS image processing pipeline that can be applied on the exemplary microscopy dataset is available in Dataset EV2. Spherical neighborhoods and their image sources can be directly inspected online at https://ibios.dkfz.de/iterstain.

**Expanded View** for this article is available online.

### Acknowledgements

We thank Sabine Aschenbrenner for support with laboratory techniques, Siegfried Winkler, Leo Burger, and Helmuth Schaar for microscopy hardware, Antonio Politi for imaging advice and ZEN black macro interface, Maria Maier for assistance with 3D renderings, Christian Dietz for continued development of KNIME image processing, and Clarissa Liesche and Joël Beaudoin for critical comments. This study was supported by the BMBF-funded e:Med program for systems medicine (PANC-STRAT, FKZ 01ZX1605A). RE acknowledges support by the Chan Zuckerberg Inititative for his contribution to the Human Cell Atlas. The authors acknowledge support by the state of Baden-Württemberg through bwHPC.

### Author contributions

LM and CC conceived the experiments and subcellular visualization strategy. LM, KJ, and MW established antibody staining and drug treatment protocols. LM developed automated iterative staining workflow, conducted experiments, and image analysis. SMK developed the interaction model and performed statistical tests. CC and RE supervised this project. LM, SMK, CC, and RE wrote the manuscript. All authors commented on the manuscript.

### Conflict of interest

The authors declare that they have no conflict of interest.

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
