## [Review Process File · Molecular Systems Biology]

Unraveling mitotic protein networks by 3D multiplexed epitope drug screening

Lorenz Maier, Stefan M. Kallenberger, Katharina Jechow, Marcel Waschow, Roland Eils and Christian Conrad

Review timeline:

Submission date:	23 rd January 2018
Editorial Decision:	6 th March 2018
Revision received:	8 th June 2018
Editorial Decision:	10 th July 2018
Revision received:	17 th July 2018
Accepted:	19 th July 2018

Editor: Maria Polychronidou

Transaction Report:

1st Editorial Decision

6th March 2018

Thank you for submitting your work to Molecular Systems Biology. We have now heard back from the three referees who agreed to evaluate your study. As you will see below, the reviewers appreciate that the presented approach seems potentially interesting. They raise however a series of concerns, which we would ask you to address in a revision of the manuscript.

Overall, the reviewers' recommendations are rather clear and I think that there is no need to repeat all the points listed below. Please let me know in case you would like to discuss any of the reviewers' comments in further detail.

REFeree REPORTS.

Reviewer #1:

Maier et al. present a technically highly challenging quantitative analysis of protein localisation in 3D cell culture spheroids. The presented method combines sequential antibody staining, with automated confocal imaging, image registration, analysis and modelling. The work is technically brilliant and impressive. It is less clear what is the overall relevance of the presented research. The authors do confirm previously describe protein localizations especially in the mitotic apparatus, they detect some quantitative differences between different cell lines and upon treatment with different drugs and propose new co-localisation hypothesis among the proteins selected for the study. However all this is very poorly motivated and this is my one and only complaint about the presented work.

The paper entirely lacks any introduction. The first paragraph essentially restates the abstract. There is no rationale presented for the massive technological development, nor any background on the biology of protein localization in mitosis and how it relates to tumorigenesis. We simply do not know why this study was performed, what were the aims. The second paragraph is even more mysterious. It appears to be prefiguring the results of the study but there are no references to Figures. The barrage of gene names begins without any motivation whatsoever.

The method itself is described extremely tersely. Considering that it is the central strong point of the paper, I believe more attention should be given to it. Figure 1 is very nice and clear as are all the Figures. The text however, does not give them justice and the reader is mostly left alone trying to figure out what data the Figures show since only a seemingly random sample of the results is discussed in the text.

In summary, I strongly believe that this is a nice work that could be published in MSB, however it has to be presented completely differently. The authors need to motivate their work and place it within the state-of-the-art. They have to describe the methods in more detail in the manuscript main text. They need to create a clear narrative of why the comparisons of mitotic components in two different spheroid lines are important, what outstanding questions are being addressed and what do the results show regarding the posed questions. Same goes for the comparisons upon drug treatment and the co-localisation analysis.

Reviewer #2:

Review Summary

This study implemented a novel pipeline and analysis of multiplexed immunofluorescence data. The analysis focused on mitotic events and mapped these to a spherical coordinate system which the authors call SpheriCell which in turn is projected to a 2D polar coordinate system for visualization. The authors first demonstrate the utility of this pipeline by comparing non-tumorigenic and tumorigenic stages of the MCF10A/MCF10CA breast cancer progression model grown in 3D spheroids. This is followed by an analysis of the same two state cell model after 48hr treatment by twelve mitotic inhibitors on twelve proteins known to be involved in mitosis, specifically spindle formation and chromatin stress. The authors examine inhibitor impact on protein expression, change in radial location of the protein (eccentricity), and change in angular orientation of expression relative to the spindle poles. Here they note that the tumorigenic MCF10CA cells seem more sensitive to spindle interference. Lastly, the authors perform an interesting and novel analysis of predicted mutual affinities for protein pairs. Through forward selection they identify 16 novel potential colocalization affinities (as measured by likelihood-ratio test). This paradigm ultimately presents an exciting and relatively high throughput method for screening specific cell types/states in organoid culture and evaluating changes in protein localization and coupling of localizations through highly multiplexed imaging.

The manuscript is clearly written and easy to understand.

Major comments

The SpheriCell coordinate system around which this analysis is based consists of "six spherical neighborhood shells [that] grow linearly in their radius from the mitotic center, and the inner four span the identified nucleus area". First, it is unclear whether the nuclear area is well defined and scaled per-cell here, or if this is just a rule of thumb observed by the authors. Please clarify. More importantly, the linearly scaling of partitions along the radial axis leads to non-uniform sector volumes. In other words, changes in mean intensity within a sector further from the center of the spheroid represents larger changes in total protein abundance compared to equal intensity changes in sectors near the center of the sphere. [Note: This is not an issue with wording. The authors correctly call their measures concentration (intensity/volume), however in some cases total abundance may be more relevant (e.g. DAPI).] This has the effect that concentration features calculated in SpheriCell are dependent on cell size as pointed out by the authors (page 4 and Fig 2a). This could be avoided if

mitotic events are morphed to a unit sphere or other canonical shape using non-rigid alignment rather than dividing intensity by sector area. One recent work did something similar and may be useful [1]. It would be interesting to determine which effects are ultimately due to difference in cell size (In Fig 3a MCF10CA vs MCF10A we see the DAPI deviation mentioned by authors as being size dependent, are other effects due to cell size change? gamma-tubulin in the same line of Fig 3a looks very similar).

To correct for this, a scale invariant analysis of protein abundance should be added. This can be done by morphing to a canonical geometry or by computing units as moles by multiplying through by sector volume. (The authors appear to correctly treat this in eq 9 when modeling localization affinity.)

The authors incubate each condition for 48hrs and evaluate effect. Effects on the relative frequency of mitotic events, distribution of mitotic phases, and mitotic cell size is not discussed (though important, see previous comment). Adding this analysis would strengthen the discussion in the paper and may offer new insights particularly surrounding the spindle fragility hypothesis posed by the authors.

This is important for a few reasons, namely:

Failed mitotic events, for example due to dysfunctional spindle assembly, may not be recognized in this pipeline by the auto-screener as mitotic at all.

A decrease in frequency of segregation events relative to metaphase events may indicate disruption of chromosome segregation.

The study seems to lack some controls necessary in determining the reliability of the results presented. Specifically:

When comparing effects of inhibitors, it would be desirable to have some technical replicates to give an estimate of the noise in the measure. This doesn't appear to have been done. Do results collected with a different set of spheroids on a different day still show the same behavior? The authors do appear to have replicate controls (Fig 1), but they are not currently used as such. At a minimum some estimate of error can be given using the multiple control experiments. Confidence intervals on the measures should be stated for these experiments or a held-out set of controls should be compared alongside other comparisons. Ideally technical replicates for at least some inhibitors would be performed.

When looking at predicted affinities, the authors used reported interactions from Ingenuity Pathway Analysis (IPA) as the ground truth. It would be interesting to determine how many of these interactions would be predicted by the model if not strictly enforced as several of these have very low mutual affinity coefficients. This could be done either by withholding some or all of the IPA reported interactions.

Minor

The authors present 3D SPECS (Spatial characterization of Protein Expression Changes by microscopic Screening) stating this "workflow encompasses iterative antibody staining of proteins, high-content imaging, and machine learning based classification of mitotic states." however it does not seem that there was an automated classification of mitotic states. As stated in the supplement (p19) "Segmented areas were manually annotated with their mitotic phase". Perhaps the authors mean automated detection of mitotic events rather than classification of mitotic states.

The major development in this study is related to the pipeline used. Central to this is the image-processing and mathematical modeling software developed (in addition to some rather novel experimental methods). Though the authors describe the development of the SpheriCell coordinate system and the subsequent protein affinity estimation, the full pipeline is not entirely clear. Adding a schematic of the KNIME pipeline with annotation as to which stages of the workflow required manual input would be useful in assessing this novel pipeline and its applicability for future labs. Though not obligatory by MSB it would be useful if this pipeline was made publicly available on a git versioning system (github, bitbucket) and/or containerized virtual machine (docker, singularity).

One of the most interesting parts of the work centers around the affinity prediction. I feel that discussion of this could be expanded somewhat in the text.

Some of the summary statistics in Figure 3 may be incomplete in describing a phenomenon, namely

the change in eccentricity. One could imagine a protein that is expressed radially uniformly equidistant from the center and boundary of the sphere prior to treatment. Under inhibitor, the protein expression is half in the outer most region and half in the center of the cell. This dramatic rearrangement would produce no shift in eccentricity as the two shifts would cancel (though this is admittedly likely very rare).

The authors use the term "compartments" to refer to their spherical coordinate system representation of the cell. Traditionally "compartments" are used to refer to subcellular localizations/organelles in the field of cellular biology (e.g. nucleoli) and while some of the "sectors" or "SpheriCell compartments" correspond to being within a given organelle (nucleus for the inner 4 rings), this nomenclature may create some confusion.

The number of mitotic events in each measure should be noted where possible. Particularly

- Fig 2 for each row.
- Fig 3 for each row split by metaphase and segregation.

Both eccentricity and orientation w.r.t. inhibitors could probably be moved to supplement as they are largely non-impacted and currently take up a large amount of space in Figure 3.

Figure 3 a should be split into MCF10CA vs MCF10A (a.1) and an inhibitor effects block (a.2). I also feel that Fig 3 c-e could be a separate figure as these deal with affinities, a different topic than the previous figure parts.

The model of protein affinity makes some steady state assumptions which seem potentially problematic given the highly non-steady-state behavior of mitosis.

Why are examples in Fig 1 manually rotated? Isn't this done automatically for the presented work?

Please comment on how parameter intervals were chosen for alpha, d and beta. Why is the range on beta so much larger?

On page 10 the authors state that SpheriCell does not require alignment of cell division in 3D. Doesn't it require 3D alignment of the spindle-poles?

Affinities reported particularly after treatment with inhibitor (Fig 3e) are restricted to only previously identified affinities. Allowing the model to re-pick affinities could uncover novel insights.

An interesting next-step for this work could be to evaluate the effect of inhibitors together to estimate affinities and simultaneously infer the mechanism of action in terms of which affinities are impacted by each drug. This is likely out of scope for the current work, but would be an interesting future direction. Supplementary figure 2 is a step towards this and interesting to note which affinities are most conserved across inhibitors.

Typographical errors:

Fig 3b is not referenced in the text (discussed top of page 6)

Supplementary Figure 1 caption needs to be corrected. Lettering is currently off and psi should only have range 1-3.

On page 30 a figure reference is missing on "Supplementary note figure" (should be 3a).

References:

[1] Cai et al. An experimental and computational framework to build a dynamic protein atlas of human cell division, bioRxiv 2017.
(<https://www.biorxiv.org/content/biorxiv/early/2017/12/01/227751.full.pdf>,
http://www.mitocheck.org/mitotic_cell_atlas/index.html)

Reviewer #3:

Summary of the manuscript:

Maier et al. used organotypic breast tissue spheroids as a platform to develop a semi-automated workflow. Using a novel process called 3D SPECS, 12 proteins within the mitotic spindle were serially immunostained and imaged in all spheroids. A machine learning approach was used to segment out all mitotic cells fixed during metaphase to anaphase for analysis. A novel analytical tool, SpheriCell, was used to build a spherical coordinate system whereby segmented mitotic cells were reoriented to a normal angle. This image processing program builds a spherical coordinate system composed of phase angles and concentric shells normalized to the orthogonal 3D geometry of spindle axis and metaphase plane. Protein abundance was determined from normalized fluorescence immunostaining densitometry. Their mathematical model relates intensity data to protein concentrations and assumes that associations between pairs of proteins or between proteins and compartments are all independent. This group was able to determine differential protein localizations within the spindle assembly affected by anti-cancer drugs. They were also able to assay the extent to which these drugs affected breast tumor spheroids versus non-cancerous spheroids. Using a combination of their immunostaining data and Qiagen's Ingenuity's Pathway Analysis (IPA), several unpublished spindle-associated protein-protein interactions were predicted. IPA also predicted drastic changes in association affinities upon exposure to a particular cancer treatment.

Major concerns:

1) Is there an example showing your approach is truly predictive and can point to a real protein-protein interaction through either biochemical or biophysical evidence?

Whether the interaction is important or not is a different question and one that is not a sticking point.

2) Please validate that the SpheriCell normalization and densitometric analysis of immunofluorescence reflects relative protein abundances.

For example, variability among antibodies and dye coupling efficiencies (for direct fluorescent labeling of primary antibodies) complicate using densitometry to determine protein abundance. Additionally, the authors only briefly mention cell size differences between the cell lines and do not explain how cell shrinkage or swelling might affect image processing and analysis. These issues may complicate the claim that PLK1 inhibition affected protein concentration but not localization. Is there epitope unmasking or other possible staining bias; or normalization bias introduced by changes in cell size due to cell line differences or drug treatment?

This technique may be justifiable, but an acknowledgement of the pros and cons as well as a justification for this method would be appreciated.

3) The stated purpose of combining computational modeling and imaging is to separate the effects of binding affinity changes and inhibitor-mediated protein localizations. The model uses a steady state concentration of each protein's binding level to compartments, then also protein to protein affinities using IPA data. The math for how the coefficients are calculated is quite clear, but it is not apparent to us what data is going into the binding parameters for compartments. Is it from mutual affinities of proteins? If so it is unclear how that works. Is it from luminescence? How is that then separated from protein-protein binding or polymer interactions?

Minor concerns:

1) Please add scale bars to each of the cell images.

2) Please explain what mutual affinities coefficients are and how they relate to a known concept -- such as K_d , if they do -- to give biologists an idea of what the coefficients may mean in reality. To our understanding, it is the inverse of the dissociation constant between the two factors, which in turn is calculated by binding and unbinding parameters. This needs to be put into better context. The authors note that a mutual binding affinity does not necessitate a functional interaction between the proteins, but this could be better explained. Though it is stated in the latter half of page 6, the example is then explained assuming that the prediction implies interaction. Use of the coefficient is an example of the difficulty experimental biologists have when extracting actionable information

about biological systems from computational models.

3) Related to the above, please explain what the error model signifies, how large actually is it when the coefficient changes by multiple orders of magnitude? Scaling factors and residuals are integrated into this calculation, but biologically do these thresholds make sense?

4) Not sure what is meant by differences between 2 phenotypes in "more physiological conditions still accessible by high-throughput screening".

5) The use of "subcellular compartments" (throughout the manuscript) to describe protein complexes should be changed. Compartments are subcellular regions spatially isolated from each other by lipid bilayer. If you mean compartments within the context of the SpheriCell analysis then please explain this.

6) Cell fate (p6, 7th line from bottom) is a term that describes a fully differentiated stem cell and confounds the point being made.

7) Please cite IPA and/or include the URL in the references.

Grammatical/format changes:

1) Page 20, line 4: change "form" to "from"

2) Page 30, last paragraph line 4 missing figure number.

Responses to Reviewer comments

Responses to the Reviewer #1:

Maier et al. present a technically highly challenging quantitative analysis of protein localisation in 3D cell culture spheroids. The presented method combines sequential antibody staining, with automated confocal imaging, image registration, analysis and modelling. The work is technically brilliant and impressive. It is less clear what is the overall relevance of the presented research. The authors do confirm previously describe protein localizations especially in the mitotic apparatus, they detect some quantitative differences between different cell lines and upon treatment with different drugs and propose new co-localisation hypothesis among the proteins selected for the study. However all this is very poorly motivated and this is my one and only complaint about the presented work.

The paper entirely lacks any introduction. The first paragraph essentially restates the abstract. There is no rationale presented for the massive technological development, nor any background on the biology of protein localization in mitosis and how it relates to tumorigenesis. We simply do not know why was this study performed, what were the aims. Second paragraphs is even more mysterious. It appears to be prefiguring the results of the study but there are no references to Figures. The barrage of gene names begins without any motivation whatsoever.

Response 1.1

We are grateful to an overall positive assessment of our methodology and agree that it was necessary to better motivate our work, introduce our approach and document our methodology. As suggested by the editor, we changed the article format to 'Method' and included paragraphs to motivate our procedure, restructured our manuscript, and focused on methodological aspects. To better introduce and motivate our approach, we completely revised main parts of the introduction section. We further explained how our approach can extend the capabilities of drug screens to comprehensively understand drug effects on cellular processes.

The revised part of the introduction section spans pp. 2-4 in the manuscript.

The method itself is described extremely tersely. Considering that it is the central strong point of the paper, I believe more attention should be given to it. Figure 1 is very nice and clear as are all the Figures. The text however, does not give them justice and the reader is mostly left alone trying to figure out what data the Figures show since only a seemingly random sample of the results is discussed in the text.

Response 1.2

We agree that the explanation of the applied methods had to be improved. We rewrote the paragraph on p. 5 that describes the definition of spherical ROIs (also visualized in new Fig EV1). A point-to-point protocol for the image processing pipeline was included in the refined subsection “*Image processing*” on pp. 18-22. Addressing your advice to improve the organization of our manuscript, we also restructured the results section to parts with subheadings.

To refine the explanation of the ROI definition, we wrote on pp. 5-6 of the revised manuscript (see also Response 2.1 to the comment of Reviewer #2):

*“[...] Usually, mitotic cells in culture divide in different orientations, which complicates comparisons between different sets of single cell data. To investigate mitosis as an example for a topographically ordered cellular process, we applied a novel representation named **SpheriCell** that facilitates spatial alignment of subcellular events by registration of a spherical coordinate to cellular landmarks. Within the defined spherical coordinate systems of the cellular space, protein concentrations are then measured in a standardized set of 3D partitions.*

For cells in metaphase, the spindle axis perpendicular to the metaphase plate was used as landmark (Fig 1B). The mitotic axis was defined by the shortest half axis of an ellipsoid fitted to the nuclear DAPI signal. Next, three sectors were delineated relative to the mitotic axis, either parallel (polar), diagonal, or in the division plane (equatorial). Six shells with equal radius intervals were centered to the nuclear ellipsoid in a way that the fourth shell was scaled to the longest half axis of the ellipsoid (Fig EV1A). For cells during segregation, the mitotic axis was specified by the line between two ellipsoids fitted to the daughter nuclei (Fig EV1B). Six equally spaced shells were defined by centering the fourth shell to the centers of the two ellipsoids. Following this procedure for cells in metaphase and segregation, outlines of cells growing in spheroids were approximated. We chose the size of the SpheriCell maps to fully cover intracellular protein distributions of the observed proteins involved in mitosis. Finally, a system of 18 spherical ROIs was created by intersecting sectors and shells. SpheriCell maps were projected on 2D planes for enhanced visualization (Fig 1B). [...]”

Furthermore, we provided a detailed point-to-point list about the image processing workflow in the subsection “*Image processing*” on pp. 18-22 that can be applied using the provided code (Dataset EV2) and an exemplary microscopy imaging dataset. During the revision process, the sample dataset is available on https://ibios.dkfz.de/documents/iterstain/MSB-18-8238_sample_dataset.zip. After acceptance of the manuscript, the contents will be made publicly available in the BioStudies database (<https://www.ebi.ac.uk/biostudies/>).

“The next steps are provided as protocol to apply the image processing workflows provided in Dataset EV2 on a sample dataset available in the BioStudies database (<https://www.ebi.ac.uk/biostudies/>).

- **Segment DAPI signal in 3D with 1 - 3d segmentation overview + storage workflow.** *It employs a region growing algorithm (Berthold et al, 2008).*
 - *Manually set seeds within mitotic positions either in 2D projections or in case of overlaps, in 3D images.*
 - *Add borders to closely neighboring nuclei, especially in z-direction.*
 - *Annotate segmented areas with their mitotic phase. The workflow then joins ana-/telophases to collect cells in segregation.*
 - *Verify assignment of segregating split chromatin regions to a single dividing cell was verified with β -Tubulin staining.*
- **Register consecutive stacks per imaging position with 2 - 3dRegWithCleanup.**
 - *This workflow provides a batch compatible subpixel alignment using Fiji (Schindelin et al, 2012), its plugin **Correct 3D Drift** (Parslow et al, 2014) including multi time scale, subpixel, and edge enhancements, and plugin **MultiStackReg** (Thevenaz et al, 1998) with scaled rotation.*

- Verify registration and annotate mitoses. If you want to create your own annotations, follow workflows **3a** through **3d**. Otherwise, proceed with workflow **3d** to use our annotations.
 - Exclude multiple mitoses per sphere (“overfilling”), and indistinct mitotic phases (“unclear”). Very early anaphases that started to segregate very recently are also to be skipped, as they would require an own class between metaphase and segregation.
 - **3a - registration tester** uses a custom virtual autofocus on DAPI that selects physically highest local maximum of variance. It antibody channels and allows to check for β -Tubulin staining strength. Exclude missing cells due to loss of Matrigel or failed registration from further analysis.
 - Refine the selection with **3b - verify registration**. It shows the central slice of the image stacks and slices 40 and 20 interpolated steps above and below.
 - Annotate indistinct mitotic phases based on their DAPI signal and antibodies against Aurora A and β -Tubulin using **3c - annotate mito areas**. Assign to metaphase or segregation if two centrosomes can be detected which are connected via β -Tubulin to the chromatin regions.
 - Collect all single annotations of registration, staining strength, and telophase pairing with **3d - combine annotations**.
- Calculate SpheriCell partitioning with workflow **4 - sphericalAnalysis**.
 - In preparation of this workflow, generate spherical segment angles with Recursive Zonal Equal Area Sphere Partitioning Toolbox (EQSP) (Leopardi, 2006) for 180 areas, or use our precomputed file **bins_180.oct**.
 - For the spherical neighborhood, the main workflow interpolates images linearly in Z to match the X/Y pixel dimensions.
 - Orientation of mitoses is identified with 3D ellipsoid fits. To this end, the KNIME workflow calls **3d_ellipsoid_fitting.ijm**, a batch compatible wrapper for calculation of ellipsoid fits using the 3D ImageJ suite (Ollion et al, 2013).
 - A custom embedded R script joins the EQSP areas to segments and to fits them in size and orientation to the individual mitoses. Metaphases could use those values as-is, but the size of segregating cells is overestimated by the ellipsoid fit and replaced by the centroid distances of their individual chromatin regions. Their 3D orientation uses the average of the first two eigenvectors and the normalized centroid to centroid vector as third. Subsequently, 3D segments were binned in three spherical intervals (equatorial, [-30°,30°] and [150°,210°]; diagonal, [30°,60°], [120°,150°], [210°,240°] and [300°,330°]; polar, [60°,120°] and [240°,300°]; see Fig 1B). The six spherical neighborhood shells grow linearly in their radius from the mitotic center, and the inner four span the identified nucleus area.
- Combine 3D bin intensities and annotations to an R (R Core Team, 2017) representation using **multiSphere.R**, which employs R packages **data.table** (Dowle & Srinivasan, 2016), **plyr** (Wickham, 2011), and **stringr** (Wickham, 2017)
- Generate SpheriCell plots for drug and antibody effects with **makePlots.R**. This script makes use of R packages **ggplot2** (Wickham, 2009), **RColorBrewer** (Neuwirth, 2014), and **Cairo** (Urbanek & Horner, 2015).
- Complete output image data with a total size of about 150GB can be inspected via **Shiny** (Chang et al, 2016) web application in **ui.R** that shows SpheriCell plots in a responsive web interface. It can be accessed at <https://ibios.dkfz-heidelberg.de/iterstain>.
 - Upon selection of a SpheriCell plot, microscopy images of corresponding treated and untreated cells are shown side-by-side, each with three layers around the central slice. Uses **RBioFormats** (Oles, 2017), **EBImage** (Pau et al, 2010), and **devtools** (Wickham & Chang, 2016).“

In summary, I strongly believe that this is a nice work that could be published in MSB, however it has to be presented completely differently. The authors need to motivate their work and place it within the state-of-the-art. They have to describe the methods in more

detail in the manuscript main text. They need to create a clear narrative of why the comparisons of mitotic components in two different spheroid lines are important, what outstanding questions are being addressed and what do the results show regarding the posed questions. Same goes for the comparisons upon drug treatment and the co-localisation analysis.

Response 1.3

We would like to thank the Reviewer here again for this feedback. Apart from feeling more comfortable with presenting our work from a technical angle, combining iterative immunostaining, automated HT microscopy in more physiological 3D cell cultures. We also want to give a strong rationale why the specific mitotic application was initially important to us. In cancer cell biology mitosis is still fundamental to understand and direct further drug development, so far with limited success due - in our belief - to technical hurdles in experimental assays. Here, the number of proteins and uncovered complexity of processes in mitosis are directly addressed in an unperceived novel way. Therefore, state-of-the-art context or references hardly exist. The comparison of most critical mitotic segregation phases in non-malignant and malignant 3D spheroids allows to identify differential drug sensitivities in the MCF10 tumorigenic progression model. Beyond this, the study is conceptually conducted as an explorative screening of protein affinities and drug inhibitions.

We addressed these issues by the new paragraphs on pp. 3-4 of the revised manuscript:

“Establishing this workflow was initially motivated by the goal of applying multiplexed staining to drug screening, to add an additional information rich layer of inhibitory processes in mitotic pathways, as several drug compounds targeting cell division unexpectedly failed in clinical trials (Chan et al, 2012; Marques et al, 2015; Otto & Sicinski, 2017). Conventionally, drug screens only account for measures as IC50 values in monolayer or, more recently, in 3D cell cultures (Jabs et al, 2017). We applied our 3D SPECS workflow to quantitatively study the differential topography of mitosis in tumorigenic MCF10CA and non-tumorigenic MCF10A cells. Experiments were performed in a 3D cell culture system regarded as physiologically more relevant than a planar cell culture because several features of these cells as differentiation, growth arrest or formation of acinar structures depend on 3D growth (Imbalzano et al, 2009). To capture the most critical events, we distinguished between cells in metaphase or cells during segregation (anaphase and telophase combined). We chose this well-established model of a non-malignant progenitor and in vitro-derived malignant cells to sensitively characterize phenotypic changes in the cellular architecture during mitosis related to malignant transformation.”

Furthermore, we added on p. 4:

“[...] Failures in these specific mitotic checkpoints can lead to disruption and catastrophe of mitosis followed by autophagic or necrotic events and therefore are investigated as potential anti-cancer drugs. The success of future mitotic checkpoint-targeted cancer therapies will depend on such complex 3D cell culture based screens to uncover synthetic lethal interaction or resistance of potent compounds in vulnerable mitotic cancer cells (Chan et al, 2012; Otto & Sicinski, 2017).”

Responses to the Reviewer #2:

Review Summary

This study implemented a novel pipeline and analysis of multiplexed immunofluorescence data. The analysis focused on mitotic events and mapped these to a spherical coordinate system which the authors call SpheriCell which in turn is projected to a 2D polar coordinate system for visualization. The authors first demonstrate the utility of this pipeline by comparing non-tumorigenic and tumorigenic stages of the

MCF10A/MCF10CA breast cancer progression model grown in 3D spheroids. This is followed by an analysis of the same two state cell model after 48hr treatment by twelve mitotic inhibitors on twelve proteins known to be involved in mitosis, specifically spindle formation and chromatin stress. The authors examine inhibitor impact on protein expression, change in radial location of the protein (eccentricity), and change in angular orientation of expression relative to the spindle poles. Here they note that the tumorigenic MCF10CA cells seem more sensitive to spindle interference. Lastly, the authors perform an interesting and novel analysis of predicted mutual affinities for protein pairs. Through forward selection they identify 16 novel potential colocalization affinities (as measured by likelihood-ratio test). This paradigm ultimately presents an exciting and relatively high throughput method for screening specific cell types/states in organoid culture and evaluating changes in protein localization and coupling of localizations through highly multiplexed imaging.

The manuscript is clearly written and easy to understand.

We thank the reviewer for an overall positive assessment of our submitted manuscript.

Major comments

The SpheriCell coordinate system around which this analysis is based consists of "six spherical neighborhood shells [that] grow linearly in their radius from the mitotic center, and the inner four span the identified nucleus area". First, it is unclear whether the nuclear area is well defined and scaled per-cell here, or if this is just a rule of thumb observed by the authors. Please clarify.

Response 2.1

We totally agree that it was necessary to describe the definition of spherical ROIs in more detail. To improve descriptions of the procedure, we rewrote the section about the SpheriCell coordinate system, created the new Fig EV1 and included a step-by-step protocol in the subsection "*Image processing*" (pp. 18-22).

At first, mitotic cells were identified by a tree ensemble classifier (see also Response 2.6) and assigned either to metaphase or segregation. For each metaphase cell, an ellipsoid was fitted to the DAPI signal (see Fig EV1A below). The shortest half axis of the ellipsoid was used to define the mitotic axis because it was perpendicular to the metaphase plate. Then, six shells with equal radius intervals were arranged in a way that the fourth shell was centered on the longest half axis. We chose the size of the SpheriCell maps to fully cover intracellular protein distributions of the observed proteins involved in mitosis. In cells during segregation, two ellipsoids were fitted to the DAPI signals of the two chromatin regions. The mitotic axis was defined by the line connecting the two ellipsoid centers. We observed that cell volumes would have been overestimated by defining two spheroids centered on the two ellipsoids. However, intracellular distributions of the measured proteins could be well covered by centering the shells on the half of the line connecting the ellipsoid centers (Fig EV1B). Then, fourth shell of six shells with equal radius intervals was scaled to the two ellipsoid centers.

Notably, this procedure resulted in only rough estimates of cell volumes and abundances, but allowed a standardized extraction of intensities in spherical ROIs. For this reason, we focused on measures proportional to concentrations that were not biased by cell volume estimates (see also Response 2.2).

We wrote on pp. 5-6 of the revised manuscript:

"[...] Usually, mitotic cells in culture divide in different orientations, which complicates comparisons between different sets of single cell data. To investigate mitosis as an example for a

topographically ordered cellular process, we applied a novel representation named **SpheriCell** that facilitates spatial alignment of subcellular events by registration of a spherical coordinate to cellular landmarks. Within the defined spherical coordinate systems of the cellular space, protein concentrations are then measured in a standardized set of 3D partitions.

For cells in metaphase, the spindle axis perpendicular to the metaphase plate was used as landmark (Fig 1B). The mitotic axis was defined by the shortest half axis of an ellipsoid fitted to the nuclear DAPI signal. Next, three sectors were delineated relative to the mitotic axis, either parallel (polar), diagonal, or in the division plane (equatorial). Six shells with equal radius intervals were centered to the nuclear ellipsoid in a way that the fourth shell was scaled to the longest half axis of the ellipsoid (Fig EV1A). For cells during segregation, the mitotic axis was specified by the line between two ellipsoids fitted to the daughter nuclei (Fig EV1B). Six equally spaced shells were defined by centering the fourth shell to the centers of the two ellipsoids. Following this procedure for cells in metaphase and segregation, outlines of cells growing in spheroids were approximated. We chose the size of the SpheriCell maps to fully cover intracellular protein distributions of the observed proteins involved in mitosis. Finally, a system of 18 spherical ROIs was created by intersecting sectors and shells. SpheriCell maps were projected on 2D planes for enhanced visualization (Fig 1B). [...]"

Furthermore, we included Fig EV1 to visualize the procedure for automatically detecting the mitotic axis:

Figure EV1 – Definition of spherical ROIs

A In cells during metaphase, a 3D ellipsoid (black area) was fitted to the DAPI signal to determine the orientation of the mitotic axis (green dotted line). This axis was defined by the smallest axis of the ellipsoid. Relative to the mitotic axis, three spherical sectors (I, polar; II, diagonal; III, equatorial) were delineated. Six shells were defined by dividing the sphere radius into six equally large intervals, of which the inner four spanned the largest ellipsoid axis nucleus area. Finally, 18 spherical 3D partitions were defined as intersections between spherical sectors and shells.

B In cells during segregation, two ellipsoids were fitted to the chromatin regions (black and grey areas). The mitotic axis (green dotted line) was defined by the centroid to centroid vector. The map of spherical ROIs was specified relative to the centroid distances of the two chromatin regions. Six shells with equal radius intervals were defined by scaling the fourth shell to the distance between the centers of the two ellipsoids (red arrows).

More importantly, the linearly scaling of partitions along the radial axis leads to non-uniform sector volumes. In other words, changes in mean intensity within a sector further from the center of the spheroid represents larger changes in total protein abundance compared to equal intensity changes in sectors near the center of the sphere. [Note: This is not an issue with wording. The authors correctly call their measures concentration (intensity/volume), however in some cases total abundance may be more relevant (e.g. DAPI).] This has the effect that concentration features calculated in SpheriCell are dependent on cell size as pointed out by the authors (page 4 and Fig 2a). This could be avoided if mitotic events are morphed to a unit sphere or other canonical shape using non-rigid alignment rather than dividing intensity by sector area. One recent work did something similar and may be useful [1]. It would be interesting to determine which effects are ultimately due to difference in cell size (In Fig 3a MCF10CA vs MCF10A we see the DAPI deviation mentioned by authors as being size dependent, are other effects due to cell size change? gamma-tubulin in the same line of Fig 3a looks very similar). To correct for this, a scale invariant analysis of protein abundance should be added. This can be done by morphing to a canonical geometry or by computing units as moles by multiplying through by sector volume. (The authors appear to correctly treat this in eq 9 when modeling localization affinity.)

Response 2.2

As recommended by the reviewer, we included an additional evaluation of abundances (Appendix Figure S1). In the previous version of the manuscript, in all cases, we had evaluated average intensities within ROIs that were assumed to be proportional to concentrations. This procedure was followed since concentrations, rather than abundances, are relevant with regard to the kinetics of biochemical reactions. Furthermore, only rough estimates of cell volumes could be obtained (see Response 2.1), which results in an additional error source when evaluating measures proportional to abundances.

In the current manuscript version, Appendix Figure S1 shows the same evaluation for abundances in comparison to effects on concentrations. Measures proportional to abundances were obtained by multiplying ROI volumes and average ROI intensity values. The calculation was included in the Appendix Supplementary Methods, together with descriptions of calculating eccentricity and orientation measures.

Furthermore, we visualized effects of cell line, mitotic phase and inhibitors on cell volume estimates as additional column in the new version of Fig 3A and B. Interestingly, only one inhibitor, Haspin, had slight effects on cell volume estimates. For all other inhibitors, observations with regard to fold changes of abundances or concentrations were similar (Appendix Figure S1, Fig 3B).

In our approach, morphing to a standard sphere, or approaches reported in the bioRxiv article by Cai *et al.*, would be incompatible with the idea of the SpheriCell approach because we were interested in comparing measures proportional to protein concentrations between groups of cells. Morphing would be advantageous for visualization of standard shapes as spatial outlines of cells. However, due to distortion of volume partitions, it is incompatible with comparisons of measures proportional to concentrations and therefore not the focus of our study.

We included on p. 9 of the revised manuscript:

“Analogous evaluations were conducted with regard to measures of abundances, obtained by weighting ROI intensities according to their volumes (Appendix Fig S1, Appendix Supplementary Methods). Only one inhibitor, Haspin, had slight effects on cell volume estimates. For all other inhibitors, observations with regard to fold changes of abundances or concentrations were similar.”

To document the procedure, we corrected on p. 9 of the Appendix Supplementary Methods:

“To analyze effects of protein intensities between cell lines, mitotic phases and inhibitor treatments, we defined measures proportional to concentrations or abundances. Furthermore, to compare spatial protein distributions, we defined characteristic measures of eccentricity and orientation.”

and

“Measures proportional to single-cell concentrations C were defined by weighting fluorescence intensities $I_{\mu\nu}$ with mitotic ROI volumes $V_{\mu\nu}$

$$C = \frac{\sum_{\mu=1}^6 \sum_{\nu=1}^3 I_{\mu\nu} V_{\mu\nu}}{\sum_{\mu=1}^6 \sum_{\nu=1}^3 V_{\mu\nu}}. \quad (1)$$

Analogously, measures proportional to abundances a of proteins were defined by

$$a = \sum_{\mu=1}^6 \sum_{\nu=1}^3 I_{\mu\nu} V_{\mu\nu}. \quad (2)$$

Moreover, we introduced a measure that describes the eccentricity of a protein distribution pattern. It was denoted as center of eccentricity r [...]

After internal discussion, we slightly corrected the applied procedure for statistical testing. Since protein concentrations in cells are log-normally distributed, we log-transformed data before conducting two-sample t-tests (now exactly specified as Welch’s t-test). This procedure was recommended by statisticians (Choi, 2016; Zhou & Gao, 1997). Correcting the procedure had marginal influence on the results.

We corrected on p. 22 of the revised manuscript:

“To test for significance of comparisons between controls and inhibitor treatments, we applied Welch’s t-tests. Since protein concentrations in cells are log-normally distributed, we log-transformed measures before statistical testing.”

We further corrected on p. 10 of the Appendix Supplementary Methods:

“Protein concentrations in cells are typically log-normally distributed. For this reason, we log-transformed measures before statistical testing (Choi, 2016; Zhou & Gao, 1997). Assuming samples with unequal variances, we performed Welch’s t-tests to statistically test for differences between conditions.”

The authors incubate each condition for 48hrs and evaluate effect. Effects on the relative frequency of mitotic events, distribution of mitotic phases, and mitotic cell size is not discussed (though important, see previous comment). Adding this analysis would strengthen the discussion in the paper and may offer new insights particularly surrounding the spindle fragility hypothesis posed by the authors. This is important for a few reasons, namely: Failed mitotic events, for example due to dysfunctional spindle assembly, may not be recognized in this pipeline by the auto-screener as mitotic at all. A decrease in frequency of segregation events relative to metaphase events may indicate disruption of chromosome segregation.

Response 2.3

We included, as described in Response 2.2, evaluations of effects on estimates of mitotic cell size. It is correct that it would have been valuable to furthermore investigate the effects of inhibitors on the frequency of mitotic events and fractions of cells in metaphase or segregation. To evaluate these fractions, it would have been, however, essential to assure that the ratios between segmented cells in metaphase and segregation were representative for the population of cells. This would have either required a randomized choice of mitoses or an evaluation of all mitoses per treatment. Furthermore, to measure mitotic fractions, it would have been required to count all cells in spheroids.

In the applied classification procedure, we simply assured correct assignment of mitotic phases, but did not access the overall fractions of mitotic and interphase cells in spheroids. In the initial step, spheroids in fields of view were defined as starting points (see new subsection *Image processing* on pp. 17-21), and detected mitotic cells were manually assigned to metaphase or segregation classes afterwards, aiming for similar counts. Therefore, fractions of cells in metaphase or segregation are not directly representative for the population of cells in spheroids. We were rather interested in comparing intracellular spatial protein distributions during the segregation process and investigate how inhibitors affect these.

However, we plotted ratios of cells in metaphase or during segregation for the two cell lines (Fig I). As expected, the fraction of cells in segregation phase was larger in MCF10CA compared to MCF10A cells, probably due to a possible segregation delay or arrest of MCF10CA cells. No cases could be observed, in which the ratio of metaphase to segregation cells was shifted in the same manner for MCF10A and MCF10CA cells. This indicates that possible strong overall inhibitor effects on fractions of cells in mitotic phases were covered due to the non-random selection procedure.

Figure I – Fractions of segmented cells in metaphase or during segregation.

A Fractions of MCF10A cells in metaphase or segregation that were not treated (ctr) or after inhibitor treatments.

B Fractions of untreated or inhibitor treated MCF10CA cells.

We included limitations with regards to fractions of cells in mitotic phases in the discussion section and added on p. 13:

“We did not analyze effects of inhibitors on fractions of cells in different mitotic phases since we did not select mitotic cells in a randomized manner. It would be, however, interesting to link effects of inhibitors on intracellular distributions of proteins involved in mitosis with effects on the duration of mitotic phases.”

The study seems to lack some controls necessary in determining the reliability of the results presented. Specifically:

When comparing effects of inhibitors, it would be desirable to have some technical replicates to give an estimate of the noise in the measure. This doesn't appear to have been done. Do results collected with a different set of spheroids on a different day still show the same behavior? The authors do appear to have replicate controls (Fig 1), but they are not currently used as such. At a minimum some estimate of error can be given using the multiple control experiments. Confidence intervals on the measures should be stated for these experiments or a held-out set of controls should be compared alongside other comparisons. Ideally technical replicates for at least some inhibitors would be performed.

Response 2.4

We agree that it was necessary to account for uncertainties of all observed effects. For this reason, we estimated confidence intervals of all observed effects described in Fig 3 by bootstrapping. In each case, measures for fold changes, eccentricity and orientation changes were calculated for 1000 bootstrap samples to obtain 95% confidence intervals. In the revised manuscript, we included a source data table for Fig 3A, B and Fig EV2 as Dataset EV1 that includes visualized measures (fold changes, eccentricity changes, orientation changes), and furthermore 95% confidence intervals as well as p-values (comparisons MCF10A vs. MCF10CA, meta vs. segregation, inhibitor treated cells vs. non-treated cells). In response to your comment below, we further included numbers of mitotic events for all comparisons (see Response 2.11).

Of note, it is an advantage of our 3D SPECS approach that it allows statistical evaluations of intracellular protein distributions between groups of cells, which is made possible by registration and fitting spheres to create a standardized map consisting of ROIs.

We included Dataset EV1 with measures visualized in Fig 3 and EV2 together with confidence intervals p-values, and numbers of mitoses. On p. 7 we wrote:

“All visualized measures, 95% confidence intervals, p-values and numbers of mitotic events are available in Dataset EV1.”

In the Appendix Supplementary Methods, we added on p. 10:

“Confidence intervals for fold changes, eccentricity changes and orientation changes were estimated by bootstrapping. We determined 95% confidence intervals from 1000 bootstrap samples.”

We became aware of a small mistake in our previous manuscript version. Bonferroni adjustment was necessary for 52 instead of 54 comparisons since 52 tests were conducted in each stained species (Fig 3A and B). Notably, by correcting for 52 instead of 54 comparisons (threshold for the p-value of $9.62 \cdot 10^{-4}$ instead of $9.26 \cdot 10^{-4}$), no additional tests were significant. We corrected the number throughout the manuscript.

When looking at predicted affinities, the authors used reported interactions from Ingenuity Pathway Analysis (IPA) as the ground truth. It would be interesting to determine how many of these interactions would be predicted by the model if not strictly enforced as several of these have very low mutual affinity coefficients. This could be done either by withholding some or all of the IPA reported interactions.

Response 2.5

Accordingly, we determined for all affinities according to literature interactions and predicted affinities the amount of $\Delta\chi^2$ by withholding interactions and refitting the model. This measure indicates, whether withholding an interaction significantly deteriorates the model fit.

All $\Delta\chi^2$ are shown in the new Appendix Figure S3. We observed that only the small fraction of four affinities according to literature interactions significantly contributed to explaining the data and marked these by squares in the new version of Fig 4B (BIRC5 \leftrightarrow γ -H2AX, β -Tubulin \leftrightarrow INCENP, Aurora A \leftrightarrow DAPI, CENP-E \leftrightarrow CENP-E).

Notably, the observation that an affinity parameter was not required for the model to explain the data does not imply absence of binding between these species but likely results due to non-identifiability as explained in the following.

For model discrimination, we tested whether point clouds of fluorescence intensity values could be fitted by linear terms dependent on ROI affinities α_{ii} , equivalent to fitting a plane, or second order terms dependent on affinities between species β_{ij} , equivalent to fitting a parabolic surface. This is analogous to fitting a line and a parabola to a point cloud in 2D. In cases where datapoints are only available in an interval where a parabola (parabolic surface) is close a line (plane), or due to noise in the dataset, no discrimination is possible.

As mentioned in the results section, a reported interaction does not necessarily imply strong binding and high affinity does not require a functional interaction. Furthermore, we would like to point out that the affinity model based on fluorescence measures of protein distributions has to be regarded as complementary to biophysical techniques as FRET. In case of the affinity model, predicted affinities can result from associations of proteins in larger protein complexes that also contain not measured species.

We added in the results section of the revised manuscript on pp. 10-11:

“We further tested, which affinities according to reported interactions between species significantly contributed to explaining the experimental dataset. To this end, affinity parameters were withdrawn and the model was refitted to determine the difference in χ^2 (Appendix Fig S3). Thereby, we found that only four affinities according to literature interactions significantly contributed to explaining the measured intensity distributions (marked by squares in Fig 4B). Notably, if an affinity parameter did not contribute to explaining the dataset, this does not imply absence of binding between these species but is likely due to non-identifiability. Erroneously rejecting an affinity parameter that might have been determined by other experimental techniques rather results from insufficient discrimination between linear and second-order terms when fitting to measurements in single-cell ROIs.”

Minor

The authors present 3D SPECS (Spatial characterization of Protein Expression Changes by microscopic Screening) stating this "workflow encompasses iterative antibody staining of proteins, high-content imaging, and machine learning based classification of mitotic

states." however it does not seem that there was an automated classification of mitotic states. As stated in the supplement (p19) "Segmented areas were manually annotated with their mitotic phase". Perhaps the authors mean automated detection of mitotic events rather than classification of mitotic states.

Response 2.6

The reviewer was correct. Machine learning was only used to detect mitoses. This was necessary since 196 stacks with 60 slides per stack had to be evaluated for each inhibitor as well as controls. After segmentation, mitotic phases were curated (to metaphase or segregation classes) which was important to ensure correct classification.

We corrected in the Abstract:

"This workflow comprises iterative antibody staining, high-content 3D imaging, and machine learning for detection of mitoses."

On p. 5, we corrected in the revised manuscript:

"Our setup (Fig 1A) uses confocal laser scanning microscopy together with automated detection of mitoses by machine learning, and a motorized in-built micro pipetting robot to comprehensively stain mitotic phases."

Furthermore, in the revised manuscript version, as described in Response 2.3, we refined the subsection "Image processing" to explain this part of our study in more detail. On p. 18 of the revised manuscript, we added:

*"To detect positions of cells, an Otsu threshold segmentation method was used to detect nucleus regions in the DAPI signal. The KNIME nodes **Global Thresholder** and **Image Segment Features** were used to extract all available image features. Based on these features, nucleus regions were classified as mitoses by applying a machine learning algorithm. For this purpose, the KNIME node **Tree Ensemble Predictor** was used after training the node **Tree Ensemble Learner** on a ground truth dataset of manually assigned mitoses. Thereby, mitoses were detected in 196 stacks per inhibitor with 60 slides per stack. The next steps are provided as protocol to apply the image processing workflows provided in Dataset EV2 on a sample dataset available in the BioStudies database (<https://www.ebi.ac.uk/biostudies/>)."*

Please note: during the review process, the sample dataset is available on https://ibios.dkfz.de/documents/iterstain/MSB-18-8238_sample_dataset.zip. After acceptance of the manuscript, the contents will be made publicly available in the BioStudies database (<https://www.ebi.ac.uk/biostudies/>).

The major development in this study is related to the pipeline used. Central to this is the image-processing and mathematical modeling software developed (in addition to some rather novel experimental methods). Though the authors describe the development of the SpheriCell coordinate system and the subsequent protein affinity estimation, the full pipeline is not entirely clear. Adding a schematic of the KNIME pipeline with annotation as to which stages of the workflow required manual input would be useful in assessing this novel pipeline and its applicability for future labs. Though not obligatory by MSB it would be useful if this pipeline was made publicly available on a git versioning system (github, bitbucket) and/or containerized virtual machine (docker, singularity).

Response 2.7

We agree with the reviewer and now made our image-processing pipeline publicly available along with the manuscript (Dataset EV2). In the Methods section, we provided a step-by-step protocol referring to

software routines included in the code directory (subsection “*Image processing*”). We further included an exemplary dataset to test the workflow that is available on https://ibios.dkfz.de/documents/iterstain/MSB-18-8238_sample_dataset.zip.

One of the most interesting parts of the work centers around the affinity prediction. I feel that discussion of this could be expanded somewhat in the text.

Response 2.8

To improve the description of the model in the main text, we included another figure panel explaining estimated affinities to spherical ROIs α_{ij} and affinities between species β_{ij} (Fig 4A). This was also beneficial for explaining the meaning of the parameters visualized in the Fig 4C and D. The part of the results section is now a separate section “Modeling intracellular distribution maps of proteins involved in mitosis”.

Some of the summary statistics in Figure 3 may be incomplete in describing a phenomenon, namely the change in eccentricity. One could imagine a protein that is expressed radially uniformly equidistant from the center and boundary of the sphere prior to treatment. Under inhibitor, the protein expression is half in the outer most region and half in the center of the cell. This dramatic rearrangement would produce no shift in eccentricity as the two shifts would cancel (though this is admittedly likely very rare).

Response 2.9

To test for this case, we included statistical evaluations for all inhibitors in the SpheriCell format as new Figures EV3 and EV4. In these figures, fold changes were indicated for ROIs with significant fold changes due to inhibitor treatments. To give an example, Fig II shows panel A of Fig EV3. We did not observe a case, in which the signal in the outer-most and inner regions decreased (increased) while the signal in intermediate regions increased (decreased).

Figure II - SpheriCell plots for inhibitor effects in MCF10A cells during metaphase.

SpheriCell plots indicating concentration fold changes for measured species in metaphase MCF10A cells treated by 12 inhibitors. Effects were visualized in ROIs with significant effects (Welch's t-tests performed for 18 ROIs followed by Bonferroni correction, $p < 0.05/18$).

The authors use the term "compartments" to refer to their spherical coordinate system representation of the cell. Traditionally "compartments" are used to refer to subcellular localizations/organelles in the field of cellular biology (e.g. nucleoli) and while some of the "sectors" or "SpheriCell compartments" correspond to being within a given organelle (nucleus for the inner 4 rings), this nomenclature may create some confusion.

Response 2.10

We agree that this was inexact. Accordingly, when referring to spherical maps, we used the term ROIs instead of compartments for segmented intracellular partial volumes defined by intersections of eccentricity shells and orientations throughout the manuscript. When describing estimations of affinities, we referred to "the content of spherical ROIs" or "affinities to mitotic ROIs".

For example, on p. 3, we replaced "localization to subcellular compartments" with "*localizations within intracellular maps consisting of spherical ROIs*".

The number of mitotic events in each measure should be noted where possible. Particularly

- Fig 2 for each row.
- Fig 3 for each row split by metaphase and segregation.

Response 2.11

Accordingly, we included numbers of mitotic events in the current version of Fig 2. In case of Fig 3, we decided to document numbers of mitoses in the tables containing the raw data visualized in Fig 3A and 3B that are provided as Dataset EV1 since including 24 numbers in Fig 3A and 48 numbers in Fig 3B would deteriorate readability.

Both eccentricity and orientation w.r.t. inhibitors could probably be moved to supplement as they are largely non-impacted and currently take up a large amount of space in Figure 3.

Response 2.12

According to this suggestion, effects of inhibitors on eccentricity and orientation measures Δr and $\Delta\phi$ were moved to Figure EV2.

In addition, we selected exemplary cases, in which changes in eccentricity or orientation were significant. For these cases, SpheriCell plots indicating ROIs with significant fold changes were shown in the new panel Fig 3C (significant change in Δr : γ -H2AX after Haspin inhibitor treatment, and INCENP after Aurora B inhibitor treatment in MCF10A cells during metaphase; significant change in $\Delta\phi$: BUB1 β and β -Tubulin after PLK1 inhibitor treatment in MCF10CA cells during metaphase). All other SpheriCell plots for inhibitor effects were included as Figs EV3 (MCF10A cells) and EV4 (MCF10CA cells).

On p. 8 of the revised manuscript, we added:

“[...] For these inhibitors, exemplary SpheriCell maps visualizing significant fold changes in ROIs are shown in Fig 3C, while SpheriCell maps of effects for all inhibitors and all measured species are presented in Figs EV3 and EV4.”

Figure 3 a should be split into MCF10CA vs MCF10A (a.1) and an inhibitor effects block (a.2). I also feel that Fig 3 c-e could be a separate figure as these deal with affinities, a different topic than the previous figure parts.

Response 2.13

Accordingly, we restructured Fig 3. Panels showing effects dependent on mitotic phase, cell type or inhibitors were shown in the revised version of Fig 3 and panels related to the model were moved into new Fig 4.

The model of protein affinity makes some steady state assumptions which seem potentially problematic given the highly non-steady-state behavior of mitosis.

Response 2.14

We agree that this steady state assumption was not well justified in the manuscript yet. In a previous study, we had determined that diffusion constants of proteins involved in mitosis typically have values around 1 to about 30 $\mu\text{m}^2/\text{s}$ (Wachsmuth *et al*, 2015). Similar values were observed for other intracellular proteins (Hinow *et al*, 2006; Vitriol *et al*, 2015). Therefore, protein concentrations spatially equilibrate in cells on the time scale of seconds or a few minutes. FCS experiments with proteins involved in mitosis had shown that binding and unbinding reactions of these proteins are generally fast compared to the

timing of mitosis (Wachsmuth *et al*, 2015). Due to these observations we would assume steady states of binding reactions.

We included on pp. 9-10 of the revised manuscript:

*“Reactions were assumed in steady state in agreement with the observation that diffusion, association and dissociation reactions of the measured species are typically fast compared to the timescale of biochemical reactions involved in mitosis (Wachsmuth *et al*, 2015)”*

Why are examples in Fig 1 manually rotated? Isn't this done automatically for the presented work?

Response 2.15

We admit that the formulation in the caption of Fig 1 was misleading. Only for visualization in this figure, microscopy images were manually rotated to vertically align mitotic axes.

Our image processing workflow automatically detects the mitotic axis in 3D to annotate spherical ROIs in a standardized manner for every cell. An automatic registration was done for alignment of iteratively recorded stack to correct slight movements between subsequent staining steps (see refined subsection *“Image processing”*). No further rotation was necessary since spherical ROIs were defined based on the geometry for each cell individually.

We corrected on p. 28 of the revised manuscript:

“For visualization, images were rotated to vertically align mitotic axes.”

Please comment on how parameter intervals were chosen for alpha, d and beta. Why is the range on beta so much larger?

Response 2.16

For initial fittings, we had defined large intervals of allowed values between 10^{-7} and 10^2 for all parameters. We observed that none of the parameters $\tilde{\alpha}_{meta,il}$, $\tilde{\alpha}_{segr,il}$, $d_{meta,i}$ and $d_{segr,i}$ touched the borders of the initially defined parameter ranges and that all of these parameters were larger than 10^{-3} whereas several of the parameters β_{ij} had smaller estimated values. To accelerate convergence in model fitting, we therefore adjusted the lower parameter boundary for $\tilde{\alpha}_{meta,il}$, $\tilde{\alpha}_{segr,il}$, $d_{meta,i}$ and $d_{segr,i}$ to 10^{-3} and verified that none of these parameter touched this lower border in model fittings. For β_{ij} , the large interval of allowed parameter values was pertained.

We corrected on p. 14 in the Appendix Supplementary Methods:

“Initially, for all estimated parameters, large intervals between 10^{-7} and 10^2 were allowed. Since none of the parameters $\tilde{\alpha}_{meta,il}$, $\tilde{\alpha}_{segr,il}$, $d_{meta,i}$ and $d_{segr,i}$ touched the lower interval boundary and all of these parameters were larger than 10^{-3} , we restricted these parameters to the interval between 10^{-3} and 10^2 to accelerate convergence in model fitting. Because estimates of several entries in β_{ij} had smaller values, intervals between 10^{-7} and 10^2 were pertained for these parameters.”

On page 10 the authors state that SpheriCell does not require alignment of cell division in 3D. Doesn't it require 3D alignment of the spindle-poles?

Response 2.17

We thank the reviewer for pointing this out. We rather wanted to mention that our approach does not require complex image processing operations as elastic registration in contrast to other image evaluation approaches. Instead, our workflow automatically detects the mitotic axis for each cell, and defines spherical ROIs dependent on orientations relative to the mitotic axis and eccentricities in a fitted sphere (see also Response 2.15).

We corrected on p. 12 of the revised manuscript:

“Morphometric image processing operations as elastic registration are not necessary because spherical ROIs are defined individually for each cell based on automatically detected mitotic axes and spherical fits.”

To provide more details about the procedure, we additionally included Fig EV1 to visualize the method for an automated detection of the mitotic axis (see also Response 2.1).

Affinities reported particularly after treatment with inhibitor (Fig 3e) are restricted to only previously identified affinities. Allowing the model to re-pick affinities could uncover novel insights.

Response 2.18

We are grateful for this suggestion. However, in our opinion, it would be not intuitive. The inhibitors that we used in this study were mostly small molecule inhibitors that targeted active centers of proteins and thereby inhibited protein activities. Inhibiting these activities might finally cause changes at the level of transcriptional responses but should, in general, rather be distinct from affinities between measured proteins. For this reason, we did not take into account that inhibitor treatments were linked to additional affinities between measured proteins.

We had restricted affinities according to the following rationale: affinities to the intracellular structures contained in spherical ROIs $\tilde{\alpha}_{meta,il}$ and $\tilde{\alpha}_{segr,il}$ were assumed to depend on mitotic phases, mainly due to the redistribution of intracellular structures, be equal for the two cell lines. However, affinities between species β_{ij} were assumed independent of mitotic phases and cell lines. One reason for these assumptions was the requirement of a sufficiently large number of experimental datapoints to estimate a comparably large number of parameters. Based on assumptions on affinity parameters, we could use experimental data of 205 untreated cells (234 measurements per cell), resulting in ~50,000 datapoints for estimating about 600 parameters (MCF10A and MCF10CA cell in metaphase or segregation fitted together). For estimating independent affinity parameters in sets of cells treated by inhibitors, the situation was worse, because we had experimental data from about 50 cells on average per inhibitor. Therefore, estimating affinities and performing a sequential model selection will be more unreliable using datasets for inhibitor treated cells.

We included in the discussion section of the revised manuscript on p. 13:

“Moreover, it might be interesting to further study model refinements related to treatment groups or investigate patterns of effects from inhibitor treatments.”

An interesting next-step for this work could be to evaluate the effect of inhibitors together to estimate affinities and simultaneously infer the mechanism of action in terms of which affinities are impacted by each drug. This is likely out of scope for the current work, but would be an interesting future direction. Supplementary figure 2 is a step towards this and interesting to note which affinities are most conserved across inhibitors.

Response 2.19

We thank the reviewer for this suggestion and agree that this would be a valuable next step. We included on p. 13 of the revised manuscript (cf. Response 2.18):

“Moreover, it might be interesting to further study model refinements related to treatment groups or investigate patterns of effects from inhibitor treatments.”

Typographical errors:

Fig 3b is not referenced in the text (discussed top of page 6) Supplementary Figure 1 caption needs to be corrected. Lettering is currently off and psi should only have range 1-3.

On page 30 a figure reference is missing on "Supplementary note figure" (should be 3a).
References:

[1] Cai et al. An experimental and computational framework to build a dynamic protein atlas of human cell division, bioRxiv 2017.

(<https://www.biorxiv.org/content/biorxiv/early/2017/12/01/227751.full.pdf>,
http://www.mitocheck.org/mitotic_cell_atlas/index.html)

Response 2.20

We thank the reviewer for pointing out typos. In the current version of the manuscript, the content of the previous version of Fig 3 was distributed to new Figs 3 and 4. In the current version of the manuscript, all parts of Figs 3 and 4 were referenced in the text.

In our opinion, the caption of previously named “Supplementary Figure 1” which is now named “*Appendix Figure S2*” was correct and consistent with labels in figure panels. To avoid incompatibilities, we converted symbols for orientations to equations (φ_1 to φ_3).

In the revised version, the figure previously named “Supplementary note figure” was now named “*Supplementary Methods Figure S3*”. We correctly referred to Supplementary Methods Figures S3A and S3B.

Responses to the Reviewer #3:

Summary of the manuscript:

Maier et al. used organotypic breast tissue spheroids as a platform to develop a semi-automated workflow. Using a novel process called 3D SPECS, 12 proteins within the mitotic spindle were serially immunostained and imaged in all spheroids. A machine learning approach was used to segment out all mitotic cells fixed during metaphase to anaphase for analysis. A novel analytical tool, SpheriCell, was used to build a spherical

coordinate system whereby segmented mitotic cells were reoriented to a normal angle. This image processing program builds a spherical coordinate system composed of phase angles and concentric shells normalized to the orthogonal 3D geometry of spindle axis and metaphase plane. Protein abundance was determined from normalized fluorescence immunostaining densitometry. Their mathematical model relates intensity data to protein concentrations and assumes that associations between pairs of proteins or between proteins and compartments are all independent. This group was able to determine differential protein localizations within the spindle assembly affected by anti-cancer drugs. They were also able to assay the extent to which these drugs affected breast tumor spheroids versus non-cancerous spheroids. Using a combination of their immunostaining data and Qiagen's Ingenuity's Pathway Analysis (IPA), several unpublished spindle-associated protein-protein interactions were predicted. IPA also predicted drastic changes in association affinities upon exposure to a particular cancer treatment.

We thank the reviewer for carefully evaluating our manuscript.

Major concerns:

1) Is there an example showing your approach is truly predictive and can point to a real protein-protein interaction through either biochemical or biophysical evidence? Whether the interaction is important or not is a different question and one that is not a sticking point.

Response 3.1

Due to reasons that we describe in the following, we had only compared additionally predicted affinities with the biochemical literature. Thereby, we found evidence for several independently predicted affinities as γ -Tubulin with CDC20 and DNA-binding of BIRC5 (Survivin). In other cases, as the predicted affinity of γ -H2AX to CDC20, or interactions with β -Tubulin or γ -Tubulin, it is known that these proteins are parts of larger multi-protein complexes.

In such cases, it will be difficult to experimentally detect affinities because experimental techniques as FRET or FCCS require that distances between proteins are in the range of only a few nanometers. An optimal technique to test for associations in larger complexes of proteins will be super-resolution microscopy, which, from our perspective, would exceed the scope of our study since we rather focused on quantifying inhibitor effects on intracellular protein distribution patterns.

Another substantial problem for experimentally detecting affinities results from the limitation of our analysis of mitotic cells. Several proteins form complexes during mitosis that are not present in other cell cycle phases. Therefore, experimental techniques that rely on population averages, in presence of a low fraction of mitotic cells, are not suitable.

We included this as a limitation of our approach in the manuscript on p. 13:

“In cases, in which associations of proteins were predicted, especially in those involving γ -H2AX, β -Tubulin or γ -Tubulin, it is likely that these proteins were present in larger multi-protein complexes. In such cases, associations might be undetectable with established biophysical techniques as FRET or FCCS because distances between proteins will exceed the required proximity. In the future, such associations in larger protein complexes might be determined by in vivo super-resolution microscopy.”

2) Please validate that the SpheriCell normalization and densitometric analysis of immunofluorescence reflects relative protein abundances. For example, variability among antibodies and dye coupling efficiencies (for direct fluorescent labeling of primary antibodies) complicate using densitometry to determine protein abundance. Additionally, the authors only briefly mention cell size differences between the cell lines and do not explain how cell shrinkage or swelling might affect image processing and analysis. These issues may complicate the claim that PLK1 inhibition affected protein concentration but not localization. Is there epitope unmasking or other possible staining bias; or normalization bias introduced by changes in cell size due to cell line differences or drug treatment?

This technique may be justifiable, but an acknowledgement of the pros and cons as well as a justification for this method would be appreciated.

Response 3.2

Accordingly, we took advantages and disadvantages of our procedure and possible error sources into account. At first, we are convinced that differences between antibody affinities to their epitopes and dye coupling efficiencies will not impact the evaluation. In Figs 2, 3, and EV2 to EV4, we eliminated possible differences in antibody affinities by reporting dimensionless parameters (fold changes, measures of eccentricity and orientation). Moreover, our model was fitted to normalized intensity measures that are independent of scales (fold changes relative to median intensities for each staining). Differences between antibodies and influences of mitotic phases on cell volumes were implicitly taken into account by including scaling factors for stainings and mitotic phases in the model that were fitted in parallel to affinity parameters (see Appendix Supplementary Methods, p. 11, equations 14 and 15).

Previous studies based on quantitative immunohistochemistry generally assumed that signals from fluorescently labeled antibodies were proportional to epitope concentrations (True, 1988). Error sources might result from steric hindrance, inhomogeneous permeation of stained tissue slices, or epitope masking dependent on the fixation method (Waters, 2009; Dapson, 2007). The general assumption of proportionality between fluorescence intensities after immunostaining and epitope concentrations was exemplarily validated by immunostaining of target proteins linked to fluorescent proteins (Mortensen & Larsson, 2001).

We tried to keep influences from staining biases small and limited error sources resulting from the technical procedure by consequent standardization of experimental procedures and automatization. Inhibitor-treated cells were only compared with untreated control cells in the same Lab-Tek that were sequentially stained in parallel. The staining procedure was standardized by automation with help of a pipetting robot controlled by software that we had developed for this purpose.

In response to a comment of Reviewer #2, we additionally evaluated effects on cell size (see Response 2.2). Most inhibitors did not take influence on estimated cell volumes. Only Haspin slightly affected the cell volume of MCF10A cells during metaphase (see current version of Fig 3B). These effects were captured by further including an analysis of abundances (Appendix Figure S1).

We included on p. 21 of the revised manuscript:

“In each SpheriCell ROI, measured fluorescence values of associated voxels were averaged. These average intensities were assumed to be proportional to protein concentrations in these ROIs in accordance with basic assumptions for quantitative immunohistochemistry and quantitative fluorescence microscopy (True, 1988; Waters, 2009). To eliminate influences resulting from differences of antibody affinities and dye coupling efficiencies, effects of mitotic phases, cell lines and inhibitors were analyzed based on scale-free magnitudes of fold changes and measures of eccentricity and orientation of protein distribution.”

3) The stated purpose of combining computational modeling and imaging is to separate the effects of binding affinity changes and inhibitor-mediated protein localizations. The model uses a steady state concentration of each protein's binding level to compartments, then also protein to protein affinities using IPA data. The math for how the coefficients are calculated is quite clear, but it is not apparent to us what data is going into the binding parameters for compartments. Is it from mutual affinities of proteins? If so it is unclear how that works. Is it from luminescence? How is that then separated from protein-protein binding or polymer interactions?

Response 3.3

We admit that the conceptual basis of the model insufficiently explained. In particular, the definition of affinity parameters (defined as the inverse of dissociation constants), had to be clarified. It was the basic concept of the model to explain spatial distributions of the stained species by (1) the recruitment to intracellular volume partitions denoted as mitotic ROIs, and (2) homo- and heterodimerization between stained species.

At the beginning of the new subsection “*Modeling intracellular distribution maps of proteins involved in mitosis*”, we included on pp. 9-10 a more detailed description about the concept of the model (additionally visualized in new Fig 4A):

“To gain a mechanistic explanation for the measured intracellular distributions, we developed a non-linear model that was calibrated with our dataset of spatially resolved fluorescence intensity measurements of proteins involved in mitosis in combination with DAPI fluorescence. The model describes concentrations for monomers, homo- and heterodimers of all measured species in spherical ROIs, defined by SpheriCell maps. In the following, we will give an overview about the model implementation and calibration (see Appendix Supplementary Methods for details). The model explains the recruitment of $i = 1 \dots 13$ measured species to $j = 1 \dots 18$ mitotic ROIs by first-order reactions with affinity parameters α_{ij} . In mitotic ROIs, all stained species can form homo- or heterodimers, described by second-order reactions with affinity parameters β_{ij} (Fig 4A). Affinity parameters were defined as the inverse of dissociation constants for recruitment to mitotic ROIs or for dimerization reactions. Of note, affinities between species were taken only into account for explaining the local enrichment of proteins but do not necessarily imply biochemical interactions between proteins. Reactions were assumed in steady state in agreement with the observation that diffusion, association and dissociation reactions of the measured species are typically fast compared to the timescale of biochemical reactions involved in mitosis (Wachsmuth et al, 2015).”

Minor concerns:

1) Please add scale bars to each of the cell images.

Response 3.4

Accordingly, we added scale bars (10 μ m) in Figs 1C and 2A.

2) Please explain what mutual affinities coefficients are and how they relate to a known concept -- such as Kd, if they do -- to give biologists an idea of what the coefficients may mean in reality. To our understanding, it is the inverse of the dissociation constant between the two factors, which in turn is calculated by binding and unbinding parameters. This needs to be put into better context. The authors note that a mutual binding affinity does not necessitate a functional interaction between the proteins, but this could be better explained. Though it is stated in the latter half of page 6, the example is then explained

assuming that the prediction implies interaction. Use of the coefficient is an example of the difficulty experimental biologists have when extracting actionable information about biological systems from computational models.

Response 3.5

We agree that this was not sufficiently explained in the results section of the manuscript. To improve this point, we refined the paragraph about the affinity model and included the new Fig 4A to better explain the estimated affinity parameters (see Response 3.3). We pointed out that estimated affinities were defined as the inverse of dissociation constants.

On p. 9 of the revised manuscript, we added:

“Affinity parameters were defined as the inverse of dissociation constants for recruitment to mitotic ROIs or for dimerization reactions.”

In the legend of Fig 4A, we wrote:

“Figure 4 - Mathematical modeling of affinities between measured species.

A Schematic graph of the mathematical model describing concentration distributions of measured species in mitotic ROIs. Spatial distributions are explained by affinities of species to cellular structures contained in mitotic ROIs α_{il} as well as homo- or heterodimeric interactions in ROIs described by affinities β_{ij} . Affinities are defined as the inverse of dissociation constants (\tilde{x}_i , unbound species i ; x_{il} , bound species i in ROI l , $x_{il} : x_{jl}$, heterodimer of species i and j in ROI l ; see Appendix Supplementary Methods for details). [...]

3) Related to the above, please explain what the error model signifies, how large actually is it when the coefficient changes by multiple orders of magnitude? Scaling factors and residuals are integrated into this calculation, but biologically do these thresholds make sense?

Response 3.6

We admit that it was beneficial to justify the applied procedure. The error model (eq. 16 in the revised Supplementary Appendix Methods) was independent on coefficients and independent on scaling factors. It was just dependent on the experimentally measured intensity values symbolized by \tilde{I}_{il} . Following the rationale of model fitting based on maximum likelihood estimation, it would be necessary to determine experimental errors of repetitive fluorescence measurements of the same protein in single cells, which is technically not well feasible. In such a case, when experimental errors cannot be exactly determined, it is recommended to use an error model, which is often used in modelling studies (Maiwald & Timmer, 2008; Kreutz et al, 2007).

In case, no error model was used, the fitting process would have ignored small intensity values (the largest fraction of datapoints) because their relative contribution to the residual sum of squares is small compared to the contribution of large intensity values.

We corrected in the Appendix Supplementary Methods on p. 13:

“To equally weight residuals for data points \tilde{I}_{il} of different magnitudes we assumed the error model

$$\varepsilon(\tilde{I}_{il}) = 0.05 \cdot \tilde{I}_{il} + 0.05 \cdot \max(\tilde{I}_{il}), \quad (16)$$

assuming that for each measurement the experimental error is given by 5% of the measurement value plus 5% of the maximal value of all included cells. This procedure is commonly recommended if repeated measurements in the same objects as single cells are not available (Kreutz et al, 2007; Maiwald & Timmer, 2008)."

4) Not sure what is meant by differences between 2 phenotypes in "more physiological conditions still accessible by high-throughput screening".

Response 3.7

We agree that this had to be better explained. The MCF10 breast cancer progression model comprises cell lines (MCF10A, MCF10AT, MCF10DCIS, MCF10CA, and others) that were sequentially derived from a common parental cell line and show increasing features associated with malignancy. These features, as invasive growth or loss of anoikis, however, cannot be observed in 2D cell cultures but only become apparent in 3D cell cultures (Imbalzano *et al*, 2009). For this reason, Imbalzano *et al*. concluded that 3D cultures of MCF10 cells were more closely related to physiological conditions.

We corrected on pp. 3-4 of the revised manuscript:

"We applied our 3D SPECS workflow to quantitatively study the differential topography of mitosis in tumorigenic MCF10CA and non-tumorigenic MCF10A cells. Experiments were performed in a 3D cell culture system regarded as physiologically more relevant than a planar cell culture because several features of these cells as differentiation, growth arrest or formation of acinar structures depend on 3D growth (Imbalzano et al, 2009)."

5) The use of "subcellular compartments" (throughout the manuscript) to describe protein complexes should be changed. Compartments are subcellular regions spatially isolated from each other by lipid bilayer. If you mean compartments within the context of the SpheriCell analysis then please explain this.

Response 3.8

We agree that this was inexact. Accordingly, we replaced the term "compartments" by - more technical - "ROIs" to describe intracellular partial volumes defined by intersections of eccentricity shells and orientations throughout the manuscript and appendix (see also Response 2.10 to a similar comment of Reviewer #2).

6) Cell fate (p6, 7th line from bottom) is a term that describes a fully differentiated stem cell and confounds the point being made.

Response 3.9

Thank you for pointing this out. In the revised manuscript, we avoided the term "cell fate" in this sentence. We corrected on p. 10:

"While we triggered DNA damage pathways with Topoisomerase II poisoning and inhibition of CHK1 (Nitiss, 2009b), activation of DNA repair mechanisms could be inferred from double strand break marker γ H2AX (Paull et al, 2000), [...]"

7) Please cite IPA and/or include the URL in the references.

Response 3.10

We included the URL in 'Materials and Methods' as well as in the new Reagents and Tools Table (MSB-18-8238_Reagents_Tools_Table.xls). Since MSB does not allow URLs in the list of references, we only referred to the URL at these two sites. Furthermore, we cited a publication related to the algorithms developed for use in IPA (Krämer *et al*, 2014) on pp. 10 and 22 of the revised manuscript.

Grammatical/format changes:

- 1) Page 20, line 4: change "form" to "from"
- 2) Page 30, last paragraph line 4 missing figure number.

Response 3.11

Thank you catching the typos, which we fixed in the revised manuscript.

References

- Chan K-S, Koh C-G & Li H-Y (2012) Mitosis-targeted anti-cancer therapies: where they stand. *Cell Death Dis.* **3**: e411
- Dapson RW (2007) Macromolecular changes caused by formalin fixation and antigen retrieval. *Biotech. Histochem. Off. Publ. Biol. Stain Comm.* **82**: 133–140
- Hinow P, Rogers CE, Barbieri CE, Pietenpol JA, Kenworthy AK & DiBenedetto E (2006) The DNA binding activity of p53 displays reaction-diffusion kinetics. *Biophys. J.* **91**: 330–342
- Imbalzano KM, Tatarkova I, Imbalzano AN & Nickerson JA (2009) Increasingly transformed MCF-10A cells have a progressively tumor-like phenotype in three-dimensional basement membrane culture. *Cancer Cell Int.* **9**: 7
- Jabs J, Zickgraf FM, Park J, Wagner S, Jiang X, Jechow K, Kleinheinz K, Toprak UH, Schneider MA, Meister M, Spaich S, Sütterlin M, Schlesner M, Trumpp A, Sprick M, Eils R & Conrad C (2017) Screening drug effects in patient-derived cancer cells links organoid responses to genome alterations. *Mol. Syst. Biol.* **13**: 955
- Krämer A, Green J, Pollard J & Tugendreich S (2014) Causal analysis approaches in Ingenuity Pathway Analysis. *Bioinforma. Oxf. Engl.* **30**: 523–530
- Kreutz C, Bartolome Rodriguez MM, Maiwald T, Seidl M, Blum HE, Mohr L & Timmer J (2007) An error model for protein quantification. *Bioinforma. Oxf. Engl.* **23**: 2747–2753
- Maiwald T & Timmer J (2008) Dynamical modeling and multi-experiment fitting with PottersWheel. *Bioinforma. Oxf. Engl.* **24**: 2037–2043
- Marques S, Fonseca J, Silva PMA & Bousbaa H (2015) Targeting the spindle assembly checkpoint for breast cancer treatment. *Curr. Cancer Drug Targets* **15**: 272–281
- Mortensen K & Larsson LI (2001) Quantitative and qualitative immunofluorescence studies of neoplastic cells transfected with a construct encoding p53-EGFP. *J. Histochem. Cytochem. Off. J. Histochem. Soc.* **49**: 1363–1367

- Nitiss JL (2009) Targeting DNA topoisomerase II in cancer chemotherapy. *Nat. Rev. Cancer* **9**: 338–350
- Otto T & Sicinski P (2017) Cell cycle proteins as promising targets in cancer therapy. *Nat. Rev. Cancer* **17**: 93–115
- Paull TT, Rogakou EP, Yamazaki V, Kirchgessner CU, Gellert M & Bonner WM (2000) A critical role for histone H2AX in recruitment of repair factors to nuclear foci after DNA damage. *Curr. Biol.* **10**: 886–895
- True LD (1988) Quantitative immunohistochemistry: a new tool for surgical pathology? *Am. J. Clin. Pathol.* **90**: 324–325
- Vitriol EA, McMillen LM, Kapustina M, Gomez SM, Vavylonis D & Zheng JQ (2015) Two functionally distinct sources of actin monomers supply the leading edge of lamellipodia. *Cell Rep.* **11**: 433–445
- Wachsmuth M, Conrad C, Bulkescher J, Koch B, Mahen R, Isokane M, Pepperkok R & Ellenberg J (2015) High-throughput fluorescence correlation spectroscopy enables analysis of proteome dynamics in living cells. *Nat. Biotechnol.* **33**: 384–389
- Waters JC (2009) Accuracy and precision in quantitative fluorescence microscopy. *J. Cell Biol.* **185**: 1135–1148

Thank you for sending us your revised manuscript. We have now heard back from the referee who was asked to evaluate your study. As you will see below, reviewer #2 is now satisfied with the performed revisions and thinks that the study is suitable for publication. S/he raises a relatively minor point, which we would ask you to address in a revision.

Before we formally accept your manuscript for publication, we would ask you to address some remaining editorial issues listed below.

REFEREE REPORTS

Reviewer #2:

I would like to thank the authors for a nicely revised manuscript. Maier et al. have performed significant revisions to strengthen their manuscript. They have gone to lengths to address all of the concerns raised in the previous review. Particularly important in my opinion are the expanded "Image processing" section and Dataset EV2 with code which give a clear impression of the procedure performed. Descriptive text and Knime workflows help in understanding and reproducibility. I was unable to access the data at the address provided (https://ibios.dkfz.de/documents/iterstain/MSB-18-8238_sample_dataset.zip) however am satisfied that the authors will make it available once published.

It was interesting to see the only inhibitor that impacted volume was Hasipn, this provides an assurance that effects seen are not due to changes in cell size.

It is unfortunate that the selection of mitotic events was non-random, however it is understandable and clear now in the paper and satisfies my question of why not analyze effect on mitotic phase distribution.

The addition of confidence intervals using bootstrapping is an adds strength to the analysis.

Another particularly interesting addition to the manuscript was made for response 2.5, where the authors found that only a small number of interactions significantly contributed to explaining the data. As the authors point out, "this does not imply absence of binding between these species but is likely due to non-identifiability", however this identifiability is still very interesting in my opinion.

The changes to figure 3 improve readability in my opinion.

In short, I believe this work can be published in MSB.

The only minor edit I suggest is that the explanation of the new dark boxes in Figure 4B should be added to the figure caption (currently only mentioned in text).

Thank you for editing our revised manuscript. We are grateful for the overall positive evaluation of our revised manuscript and suggestions for corrections.

Below, we listed responses to all comments and amendments in the manuscript. In the following, editor and reviewer comments are formatted in grey font as indented; responses are marked in blue font, *corresponding changes of the manuscript are shown in italic font*. Page numbers refer to the manuscript with accepted changes.

Reviewer #2 comment

The only minor edit I suggest is that the explanation of the new dark boxes in Figure 4B should be added to the figure caption (currently only mentioned in text).

We thank the reviewer for a positive evaluation of our revised manuscript. According to the comment, we added a description to the caption of Fig 4B (p. 39):

“Known affinities that significantly contributed to explaining the measured intensity distributions were marked by black squares.”

Corresponding Author Name: Christian Conrad

Manuscript Number: MSB-18-8238